# Groundwater-Surface Water Interactions across an Arid River Basin: Spatial Patterns Revealed by Stable Isotopes and Hydrochemistry

Liheng Wang[1,2], Yuejia Sun[1,2], Chun Yang[3] Yanhui Dong[1,2]

[1] State Key Laboratory of Deep Petroleum Intelligent Exploration and Development, Institute of Geology and Geophysics, Chinese Academy of Sciences, Beijing 100029, China
[2] College of Earth and Planetary Sciences, University of Chinese Academy of Sciences, Beijing 100049, China
[3] School of Geophysics and Information Technology, China University of Geosciences (Beijing), Beijing 100083, China

*Correspondence to*: Yanhui Dong (dongyh@mail.iggcas.ac.cn)

**Abstract.** A comprehensive understanding of groundwater-surface water (GW–SW) interactions is essential for managing water resources in arid regions, where hydrological processes are highly sensitive to climate variability and human activity. This study investigates spatial variations in GW–SW relationships across the Shule River Basin (SRB) in Northwest China, based on hydrochemical and stable isotopic analyses of 31 river water and groundwater samples. Isotopic results reveal a clear altitude effect in river water, with $\delta^{18}O$ values decreasing at a rate of $-0.08‰/100$ m, which is lower than the rate observed in the adjacent Qinghai-Tibet Plateau. In the upper reaches, river water is mainly derived from precipitation, glacier meltwater, and groundwater. In the midstream area, river water recharges groundwater at higher elevations, while spring discharge contributes groundwater back to the river at lower altitudes. In the lower reaches, irrigation return flow becomes a key recharge source for shallow groundwater. Hydrochemical results show progressive salinization along the flow path. River water TDS increases from 371.40 mg/L upstream to 1072.13 mg/L downstream, while groundwater TDS ranges from 506.51 to 1499.65 mg/L. River water is primarily influenced by silicate and carbonate weathering, whereas groundwater chemistry is governed by mineral dissolution and cation exchange reactions. These findings highlight strong spatial heterogeneity in water quality and GW–SW interactions. A conceptual model of the basin-scale hydrological cycle is proposed based on the above understanding. This model not only provides important insights into typical river–groundwater systems in arid regions of Northwest China but also serves as a valuable reference for analogous studies and the sustainable management of water resources in arid regions worldwide.

## 1 Introduction

Groundwater and surface water (GW–SW) interactions constitute a pivotal yet intricate component of the hydrological cycle, jointly sustaining ecological integrity and human water demands across diverse landscapes (Ma et al., 2024; Kuang et al., 2024). Through mechanisms such as seepage, bank infiltration, hyporheic exchange, aquifer overflow, and river leakage, bidirectional exchanges transpire continuously across nested spatial scales ranging from local riparian zones to entire basins (Kalbus et al., 2006). However, the inherent spatial–temporal variability and complex geological and hydrological controls

render the precise identification, quantification, and temporal delineation of these exchange hotspots exceedingly challenging (Sophocleous, 2002). Addressing this knowledge gap is critical for elucidating watershed water balances, interpreting system evolution under environmental stressors, and informing sustainable water resource management and ecosystem protection strategies. Consequently, advancing our mechanistic understanding of GW–SW coupling remains a focal research priority within the hydrological community.

In this context, delineating GW–SW interactions in arid and semi-arid regions is particularly critical (Li et al., 2024; Wang et al., 2024a; Zafarmomen et al., 2024). These landscapes, which cover more than 40% of the global land area, support approximately 38% of the world's population (Wang et al., 2024a), are increasing burdened by the dual pressures of climate changes and intensified anthropogenic activities, such as increasing population numbers, expanding areas of irrigated agriculture, and growing industrial demands. Under these circumstances, sustainable development and sustainable development and management of scarce water resources face unprecedented challenges (Crosbie et al., 2023). However, traditional hydrological survey and instrumentation remain difficult to apply the basin scale due to extreme climatological conditions and technical constraints (Kalbus et al., 2006). For example, seepage meters yield only point-scale estimates of exchange fluxes (Murdoch and Kelly, 2003), thermal tracing is limited to short river reaches (Banks et al., 2022), and mass-balance approaches require numerous parameters or incur large errors under strong evaporation-driven water-table fluctuations. To overcome these methodological limitations, analyses of hydrochemical composition and environmental isotopic signatures have emerged as powerful tools for characterizing groundwater–surface water exchanges across diverse spatial and temporal scales, particularly in arid and semi-arid regions (Yang et al., 2021; Zhang et al., 2023).

GW–SW interactions comprise two fundamental processes: (a) effluent conditions (gaining streams), where groundwater discharges into the river, and (b) influent conditions (losing streams), where surface water infiltrates to recharge aquifers (Sophocleous, 2002). These exchanges vary spatially and temporally, as rivers may gain water in some reaches while losing it in others, and seasonal fluctuations further modulate these fluxes (Keery et al., 2007). Numerous studies have shown that integrating stable isotopic tracers (e.g., $^2$H or D, $^{18}$O) with hydrochemical analyses at laboratory, reach, and regional scales effectively elucidates these dynamics (Aravena et al., 2024; Xie et al., 2024; Xiao et al., 2024). Due to highly intense evaporation, pronounced isotopic fractionation renders stable isotopes particularly well suited for tracing exchange processes in arid regions (Jasechko, 2019). Consequently, these methods are routinely used to quantify groundwater contributions to baseflow (Kebede et al., 2017) and to estimate the proportion of surface water used for irrigation that recharges aquifers (Wang et al., 2016). While single tracers may not fully capture the complexity of these exchanges, multi-tracer approaches yield more robust characterizations (Gómez-Alday et al., 2022). Moreover, integrating hydrochemical data further clarifies flow pathways and assesses regional water quality, thereby informing sustainable resource allocation and management (Oyarzún et al., 2016; Wang et al., 2024a). However, research on GW–SW interactions at the basin scale remains extremely scarce at present, which has consequently attracted broader attention, particularly in arid regions (Xiao et al., 2024; Wang et al., 2024b).

The Shule River basin (SRB), located in the hyper-arid and arid northwest region of China, is not only historically significant as part of the ancient Silk Road but also renowned for its numerous oases that serve as the basis of local livelihoods and

economic development (Wang et al., 2015). Therefore, regional water resources are not only vital for sustaining local socio-economic growth but also crucial for safeguarding the delicate ecosystems. The upper reaches of the Shule River lie within an ecologically fragile sector of the Qilian Mountains along the northern margin of the Qinghai-Tibet Plateau. In this headwater region, Zhou et al. (2015) employed isotopic tracers of rain, glacial melt, groundwater and river water to initially demonstrate that groundwater is the main source of river recharge. Similarly, Wang et al. (2016) also utilized an isotopic approach to

characterise the GW–SW interaction in the lower reaches of the Shule River, and concluded that agricultural irrigation water from the river is an important source of groundwater. In addition, Xie et al. (2024) meticulously examined the hydrochemistry and multi-isotope composition of river and groundwater within the SRB, and preliminarily identified the sources of substances in the water. Recognizing the importance of water resources in this region, extensive studies have been conducted through various approaches and methodologies over the years. While previous research has focused on specific or isolated agricultural

irrigation areas, others have targeted the origins and evolution of groundwater (Guo et al., 2015; He et al., 2015; Wang et al., 2015). Nevertheless, it must be emphasized that comprehensive basin-scale investigations are urgently to clarify the process of GW–SW interactions, elucidate their spatial heterogeneity, and develop robust conceptual models. Systematically characterization of water quality characteristics within this framework will not only advance theoretical understanding of hydrological cycles and offer transferable insights for analogous studies in other arid regions, but also provide a scientific basis

for the sustainable development and management of regional water resources.

Consequently, this study pursues three interrelated objectives through systematic analysis of hydrochemical composition and isotopic signatures of river water in mainstream and the groundwater samples from the SRB: 1) to characterize the spatial variability of isotopic and hydrochemical compositions of both river water and groundwater; 2) to develop a conceptual model of GW–SW interactions at basin scale compare their spatial heterogeneity; and 3) to synthesize existing hydrochemical data

and conduct a comprehensive water quality assessment to support agricultural irrigation and potable use by human and livestock. By integrating these objectives, the research will deepen theoretical understanding and provide a robust scientific foundation for sustainable water-resource management in the SRB, thereby offering transferable insights and making a meaningful contribution to the understanding of hydrological processes in arid regions worldwide.

## 2 Materials and methods

### 2.1 Study area description

The SRB is geographically located on the northern edge of the Qinghai-Tibet Plateau, in northwestern China. It is bordered by longitude 92°11′ to 98°30′ E and latitude 38°00′ to 42°48′ N. The mainstream of the Shule River is more than 620 km, covering a drainage area exceeding 40,000 km$^2$. Its mainstream origin can be traced back to a network of 347 glaciers nestled in the western section of the Qilian Mountains. Flowing in a northwest direction, the mainstream of the Shule River meanders through

many gorges in the Qilian Mountains (Fig. 1), before flowing into the Changma basin. After the Shule River flows out of the Qilian Mountains at the Changma Reservoir, it flows from east to west along the edge of the alluvial fan through Yumen, then

enters the Shuangta Reservoir, continues to flow west through Guazhou, and finally disappears in the Kumtag Desert. Therefore, it can be divided into three distinct segments, each characterized by its unique geographical features. The upper reaches, stretching from the river's source to the outlet of the Changma Reservoir (Fig. 1), lie at elevations ranging from 2080 to 5808 m. This region is primarily dominated by towering and precipitous mountain formations, interspersed with relatively gentle valleys, particularly in the vicinity of Changma. Progressing further downstream, the middle reaches encompass the vast expanse of the Yumen Alluvial Fan Plain (Fig. 1). With elevations fluctuating between 1310 to 2050 m, this segment of the river exhibits a notable decrease in altitude from south to north. Finally, the lower reaches of the Shule River encompass the extensive Guazhou Plain, abutting the northern mountainous region of the arid Gobi Desert, with elevations varying from 1020 to 1650 m. Therefore, the upstream region is primarily located in the high-altitude Qilian Mountains, while the midstream and downstream regions are situated in the relatively flat terrain of the Hexi Corridor (Fig. 1).

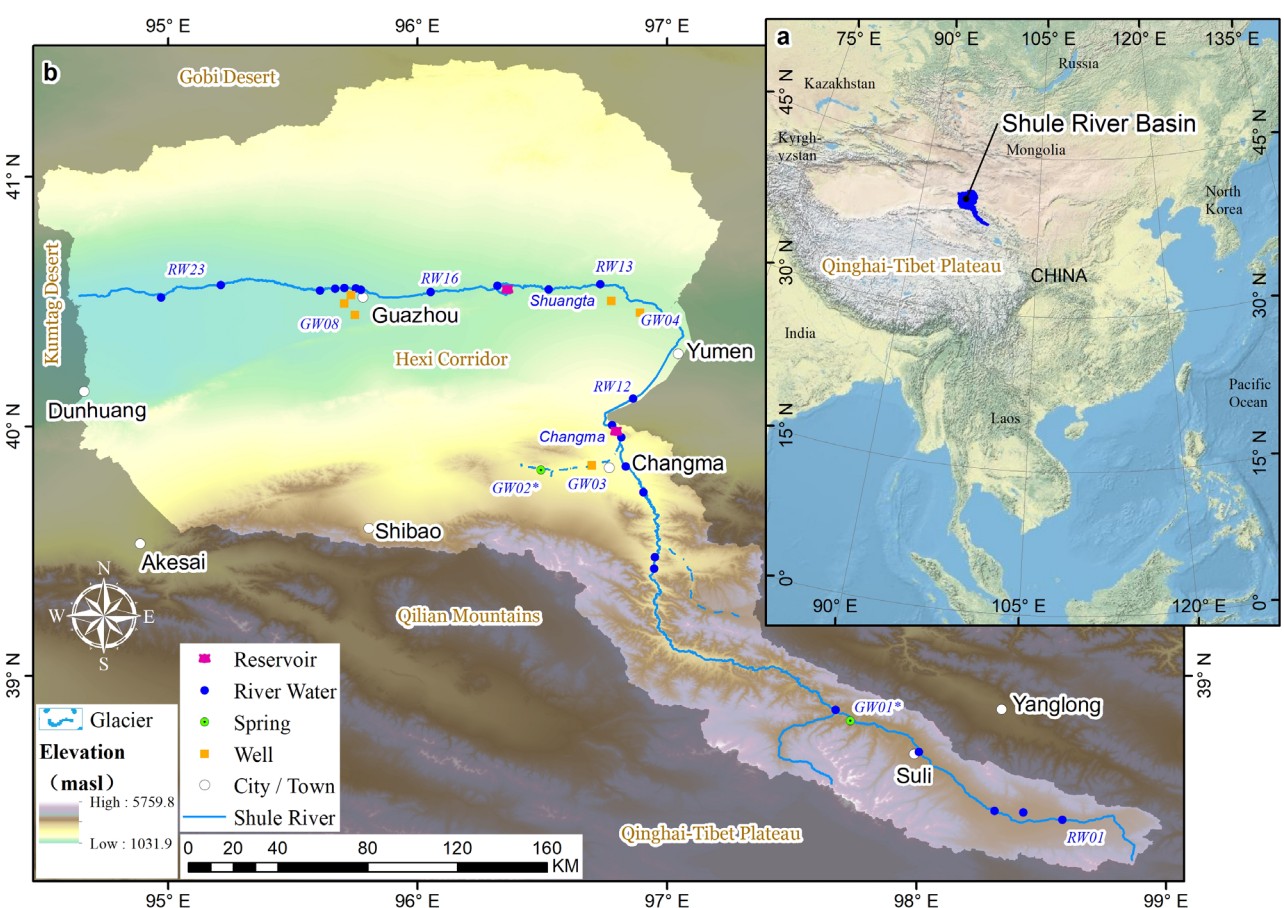

**Figure 1: (a) Geographic location of the study area (Data Source: USGS). (b) Sampling sites for river water, groundwater across Shule River Basin. The digital elevation model (DEM) used in this study was derived from the Shuttle Radar Topography Mission (SRTM) 3 arc-second (90 m) dataset (Farr et al., 2007), openly available from NASA's Earth data portal (https://earthdata.nasa.gov/)**

As for the sources of recharge for the Shule River's flow, they encompass various crucial elements, such as the meltwater from glaciers, groundwater reserves, and localized precipitation. However, the average annual runoff of the mainstream, which amounts to $10.31 \times 10^8$ m$^3$, undergoes a gradual decline as the river enters the Hexi Corridor. This decline is largely attributed to the heightened exploitation and utilization of water resources due to agricultural irrigation and industrial production, leading to a higher rate of evaporation and seepage losses.

The study area is situated in the heartland of the Eurasian continent, far from the influence of oceans, resulting in an extremely dry climate. The region is characterized by scarce precipitation and intense evaporation. The SRB, on average, receives a meager annual precipitation of 78.5 mm, while the evaporation rate is remarkably high at 3042 mm. The annual average temperature ranges from 6.9 to 8.8 °C. The southern part of the study area, which encompasses the Qilian Mountain range, belongs to a high-altitude semi-arid climate zone, with precipitation levels ranging from 100 to 200 mm and occasionally reaching up to 400 mm. The temperatures in this area remain cold, hovering between 0 to 4 °C, with an annual evaporation of about 1700mm. Conversely, the middle and lower reaches of the basin fall within a temperate arid zone, receiving an even more limited annual precipitation ranging from 36 to 63 mm. Precipitation in the study area shows pronounced intra-annual variability. More than 75 % of the annual rainfall occurs between May and September, and these events can even generate temporary floods and intermittent streams (Guo et al., 2015). The average temperatures here range from 6 to 8 °C, and the evaporation rates vary between 1500 to 2500 mm annually.

## 2.2 Geology and hydrogeology setting

The study area is situated within a highly tectonically active region in the northern part of the Qinghai-Tibet Plateau. It is characterized by intense geological processes, including thrust faults and structural uplift. Qilian Mountains, which has experienced significant uplift since the late Paleozoic, it is subjected to a prominent NNE-directed compressional tectonic force (Lin et al., 2022; Yang et al., 2020). This force has led to the formation of a series of NNE trending faults (Fig. 2), which play a crucial role in shaping the development of the Shule River system.

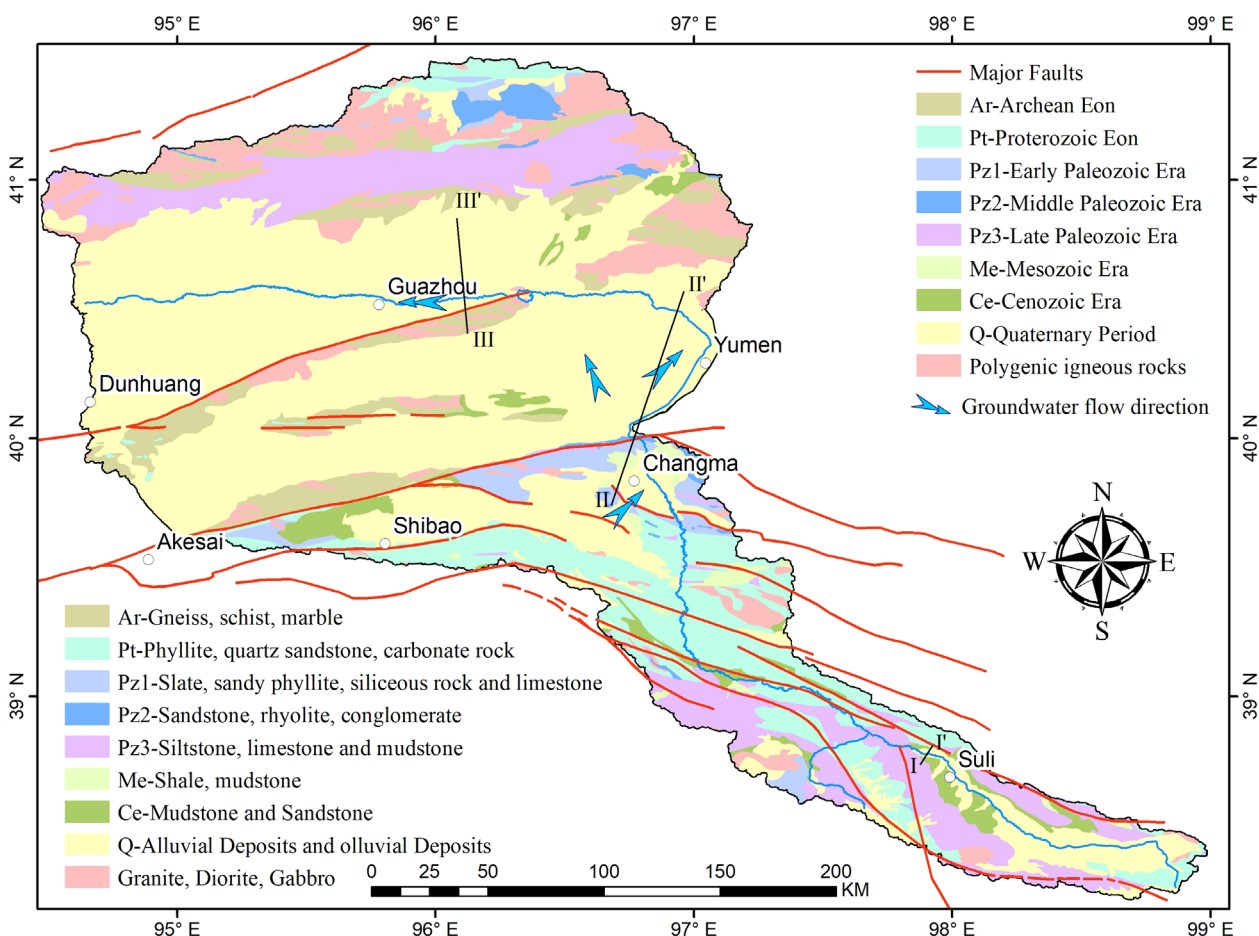

**Figure 2: Geological-stratigraphic distribution, structural framework, and hydrogeological sketch map of the study area (modified from the 1:1,000,000 Geological Map of Gansu Province, Gansu Geology Survey 1978~1980).**

The upstream region of the Shule River is extensively covered with various rock types (metamorphic, sedimentary rocks) spanning multiple geological periods (Fang et al., 2005). This includes Precambrian metamorphic rocks such as gneiss, schist, dolomite, and quartzite, as well as early Paleozoic metamorphic rocks like slate, sandstone, conglomerate, and carbonate rocks. Additionally, sedimentary rocks and granitic formations from different geological stages are also present. Furthermore, in the upstream area, sedimentary rocks dating back to the Mesozoic and Late Paleozoic periods, including sandstones, mudstones, and limestones, can be observed. Throughout the Mesozoic era, the Hexi Corridor gradually took shape and experienced significant subsidence (Meng et al., 2020). As it entered the Cenozoic era, the Qilian Mountains region underwent extensive erosion. Rivers transported substantial quantities of debris into the Hexi Corridor, leading to the deposition of thick Quaternary sediments. Over time, this process gave rise to various-sized alluvial fans. Quaternary sediments constitute a significant component of the stratigraphy in the foreland alluvial fans and basins of the study area. These unconsolidated fluvial deposits, which include loess, gravel, and sand, can reach a thickness of 500 to 600 meters in the middle to upper reaches of the Shule

River. Examples of such deposits can be observed in the Changma alluvial fan and the Yumen alluvial fan in the upper and middle reaches, respectively (Wang et al., 2017; Guo et al., 2015).

Groundwater with water supply significance in the study area is distributed in the alluvial fan plain area or on both sides of the river valley. These groundwaters exist in the pore of the Quaternary loose sediments. Previous studies have indicated that the primary source of recharge for groundwater is atmospheric precipitation, and its flow direction is intimately tied to the local topography (Fig. 2) (Guo et al., 2015). Because river water and groundwater have undergone many mutual transformations from upstream to downstream, it is believed that there is a close hydraulic connection between the two (Wang et al., 2016; Xie et al., 2022). In the upper reaches, because it is mostly mountains and canyons, the terrain changes very drastically, so scattered springs are found. In the alluvial-diluvial fan edge area, groundwater flow is impeded, giving rise to the emergence of artesian springs. This phenomenon is frequently observed in the peripheries of the Changma and Yumen alluvial fan plains. Similar to other foreland alluvial fans, the sediments are coarse-grained at the mouths of the Changma and Yumen fans but become relatively fine-grained at the edges. This depositional pattern significantly influences the spatial distribution of groundwater within these alluvial fans. In particular, in both Changma and Yumen, the aquifer systems transition from a single unconfined layer in the southern regions to multiple, interlayered confined layers towards the northern regions. Therefore, the thickness of the aquifers also varies significantly in space, ranging from tens to hundreds of meters. Correspondingly, the groundwater depth also varies, ranging from several meters to several hundred meters (Fig. 3) (Wang et al., 2016; Wang et al., 2015).

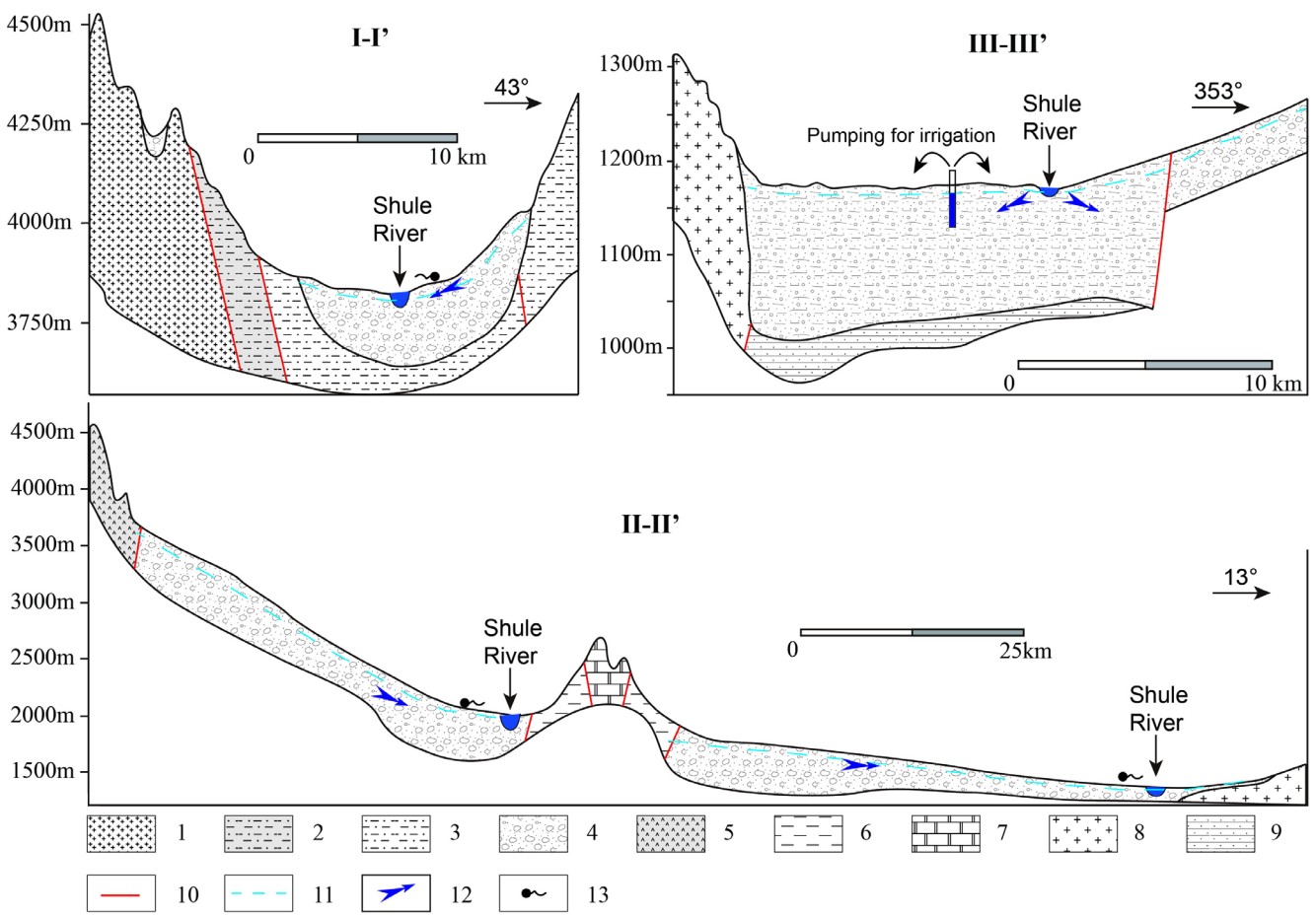

**Figure 3: Hydrogeological cross-sections of upper, middle and lower reaches of the Shule River, and their location were indicated in Fig. 2. These cross-sections were adapted from the regional hydrogeological atlas of the study area (Gansu Geology Survey 1978~1980). 1. Gneissic migmatitic granite; 2. Feldspathic quartz sandstone; 3. Sandstone; 4. Gravels, pebbles, and sand; 5. Marble and slate; 6. Slate and sandstone; 7. Carbonate; 8. Granite; 9. Mudstone; 10. Fault; 11. Groundwater table; 12. Groundwater flow direction; 13. Spring.**

## 2.3 Sample collection and analysis

Given the SRB's vast spatial extent, exhaustive sampling of groundwater and high-resolution river water at every location is neither feasible nor necessary. Accordingly, we first compiled and critically reviewed all available hydrochemical and isotopic data for precipitation, glacier meltwater, surface water, and groundwater within the SRB (Table S1). Building on this comprehensive dataset and carefully evaluating the spatial representativeness and hydrochemical–isotopic variability of existing samples, we designed our own field campaign to fill key gaps and target the most informative locations. In July 2022, a total of 31 new water sample sets were collected within the study area, whose sites were selected to capture the full range of hydrological settings (Fig.1). These samples consisted of 23 river water samples and 8 groundwater samples (including both spring and well water). To elucidate GW–SW interactions, groundwater samples were deliberately positioned 500 – 1 000 m

from the mainstem of the Shule River, ensuring that our sampling network robustly represents both regional flow paths and local exchange processes.

For river water sample collection, efforts were made to select locations situated away from the riverbed, and samples were collected from a depth of 10 cm below the water surface at flowing sections of the river. When conducting groundwater sampling, spring water samples are typically collected directly from the spring source, capturing fresh groundwater immediately upon its emergence. Well water samples are obtained from agricultural irrigation wells within the study area. To ensure the representativeness of these samples, the wells were generally pumped for three times the well volume before sample collection. According to field investigations conducted during sampling, the depths of these agricultural irrigation wells range from 80 meters to 200 meters. To enhance the water supply capacity of the wells, intake screens are installed along their entire lengths, so the samples represent a mixture of shallow and deep groundwater. All sample collection was conducted in a single round.

All samples underwent on-site filtration using 0.45-micron cellulose acetate filter membranes. They were then carefully stored in pre-cleaned polyethylene bottles, which had been rinsed three times with water from the respective sampling site. Three bottles were collected for each sample: one was acidified to a pH below 2 for cation testing, while the other two were designated for anion and stable isotope analysis. Throughout transportation to the laboratory, the water samples were stored at a constant temperature of 4 °C.

On-site measurements included pH, temperature, TDS (HANNA HI 9811-5), latitude and longitude coordinates of the sampling locations, as well as elevation. $HCO_3^-$ was determined using a titration method, while all other cations, anions, and stable isotopes were comprehensively analyzed by the Weathering and Hydro-geochemistry Laboratory at the Institute of Geology and Geophysics, Chinese Academy of Sciences. The stable isotopes (D, $^2H$ and $^{18}O$) in the water samples were analyzed using an L1102-I (Picarro, USA) and the results were expressed as delta values, defined as the per mil deviation from Vienna Standard Mean Ocean Water, as shown in Eq. 1. For precision, each sample is tested six times, with the first two measurements discarded, and the average of the remaining four used as the final value.

$$\delta = \frac{R_{sample} - R_{standard}}{R_{standard}} \times 1000 , \tag{1}$$

where R represents the isotopes ratios ($^2H/^1H$ or $^{18}O/^{16}O$) of the sample and standard, respectively. The deuterium excess value can be calculated according to (Dansgaard, 1964):

$$\text{d-excess} = \delta D - 8 \times \delta^{18}O, \tag{2}$$

**2.4 Water quality for drinking and irrigation evaluation**

The study area is a major agricultural irrigation region in the northwest of China, known as the part of the Hexi Corridor. Both surface water and groundwater play a crucial role in agricultural irrigation and drinking water. In order to assess the quality of

irrigation water versus drinking water, Sodium Adsorption Ratio (SAR), Sodium Percentage (Na%) and the total hardness (TH) was calculated. The calculation methods can be found in Wang et al. (2024a).

## 3 Results

### 3.1 Hydrochemical characteristics and evolution processes of river water

The physicochemical data of river water and groundwater in the study area is reported in Table 1. The river water exhibits weak alkalinity throughout its entire course, with an average of 8.36. The TDS in the river gradually increases as it flows downstream, with the headwater area recording the lowest TDS at 209 mg/L and the downstream area peaking at 1672 mg/L (Fig. 4). It significantly exceeds the global average TDS for rivers (115 mg/L). Based on the definition of water hardness, the upper reaches of the river (above the Changma Reservoir, Fig. 1) fall within the categories of slightly hard (150 mg/L <TH <300mg/L) and hard water (300 mg/L <TH <450mg/L), while the middle reaches (from Changma Reservoir to Shuangta Reservoir, Fig. 1) qualify as hard water. The lower reaches of the river (below Shuangta Reservoir, Fig. 1) are classified as very hard water (TH >450 mg/L).

Table 1 Basic physical and chemical data for river water and groundwater samples in the Shule River Basin. 'RW' indicate river water samples, while 'GW' stand for groundwater samples.

| No. | Temp | pH | TDS | $Ca^{2+}$ | $K^+$ | $Mg^{2+}$ | $Na^+$ | $Cl^-$ | $HCO_3^-$ | $SO_4^{2-}$ | $NO_3^-$ | $SiO_2$ | $\delta D$ | $\delta^{18}O$ | d-excess |
|------|------|------|--------|--------|------|--------|-------|-------|--------|--------|------|-------|--------|--------|--------|
| | (℃) | (–) | (mg/L) | (mg/L) | (mg/L) | (mg/L) | (mg/L) | (mg/L) | (mg/L) | (mg/L) | (mg/L) | (mg/L) | (‰) | (‰) | (‰) |
| RW01 | 5.4 | 8.22 | 209.00 | 51.00 | 1.97 | 7.70 | 16.60 | 12.72 | 168.73 | 39.69 | 1.80 | 4.67 | -58.25 | -9.41 | 17.03 |
| RW02 | 5.2 | 8.47 | 225.00 | 57.16 | 1.06 | 9.95 | 19.64 | 14.52 | 175.38 | 46.31 | 3.84 | 4.54 | -59.62 | -9.24 | 14.30 |
| RW03 | 4.8 | 8.35 | 265.00 | 64.74 | 1.73 | 8.47 | 19.34 | 26.49 | 186.40 | 51.74 | 3.78 | 4.33 | -60.41 | -9.26 | 13.67 |
| RW04 | 6.6 | 8.41 | 319.00 | 77.89 | 1.68 | 11.86 | 21.62 | 31.11 | 196.73 | 77.06 | 3.60 | 4.12 | -58.38 | -8.94 | 13.14 |
| RW05 | 6.1 | 8.56 | 365.00 | 83.47 | 1.27 | 20.45 | 28.08 | 38.71 | 225.20 | 98.55 | 3.47 | 4.29 | -61.49 | -9.32 | 13.07 |
| RW06 | 6.0 | 8.21 | 396.00 | 85.80 | 1.35 | 25.90 | 21.40 | 47.93 | 239.88 | 102.14 | 4.53 | 4.37 | -62.98 | -9.12 | 9.98 |
| RW07 | 6.3 | 8.50 | 492.00 | 86.70 | 1.22 | 26.30 | 27.40 | 51.35 | 232.05 | 144.00 | 5.27 | 4.63 | -60.79 | -9.46 | 14.93 |
| RW08 | 5.9 | 8.08 | 435.00 | 94.20 | 1.25 | 28.80 | 23.50 | 49.24 | 248.41 | 113.67 | 4.77 | 4.50 | -61.25 | -9.81 | 17.23 |
| RW09 | 5.7 | 8.01 | 496.00 | 85.80 | 1.42 | 48.50 | 12.40 | 64.36 | 221.64 | 147.47 | 5.33 | 4.92 | -78.02 | -12.12 | 18.96 |
| RW10 | 6.1 | 8.31 | 512.00 | 92.90 | 1.90 | 53.30 | 19.70 | 65.80 | 229.00 | 173.30 | 2.90 | 4.11 | -60.90 | -8.70 | 8.70 |
| RW11 | 6.7 | 8.45 | 492.00 | 96.70 | 2.17 | 30.40 | 29.50 | 41.26 | 196.75 | 178.83 | 5.70 | 4.79 | -58.19 | -8.88 | 12.85 |
| RW12 | 7.1 | 8.52 | 498.60 | 93.20 | 2.00 | 37.50 | 29.70 | 70.80 | 219.20 | 135.00 | 2.90 | 4.24 | -61.70 | -9.10 | 11.10 |
| RW13 | 7.8 | 8.37 | 542.60 | 95.00 | 2.60 | 36.40 | 46.50 | 61.60 | 195.60 | 205.00 | 1.70 | 4.56 | -53.30 | -8.40 | 13.90 |
| RW14 | 7.4 | 8.14 | 545.10 | 97.70 | 3.29 | 36.20 | 41.80 | 65.13 | 173.83 | 217.10 | 6.01 | 4.71 | -52.35 | -7.79 | 9.97 |
| RW15 | 7.4 | 8.14 | 579.00 | 101.50 | 3.72 | 41.50 | 48.80 | 72.52 | 225.21 | 204.25 | 4.53 | 4.54 | -55.54 | -8.53 | 12.71 |
| RW16 | 7.8 | 8.52 | 598.00 | 118.10 | 3.18 | 34.50 | 39.70 | 67.93 | 239.93 | 207.98 | 4.90 | 4.50 | -49.67 | -6.82 | 4.89 |
| RW17 | 8.0 | 8.22 | 650.20 | 126.00 | 3.08 | 40.30 | 56.90 | 84.80 | 258.69 | 232.96 | 3.41 | 4.63 | -50.29 | -7.14 | 6.83 |

| | | | | | | | | | | | | | | |
|---|---|---|---|---|---|---|---|---|---|---|---|---|---|---|
| RW18 | 11.0 | 8.32 | 961.00 | 169.00 | 2.95 | 76.80 | 91.00 | 155.57 | 270.86 | 410.65 | 1.10 | 4.33 | -60.70 | -8.48 | 7.14 |
| RW19 | 11.5 | 8.26 | 782.40 | 137.90 | 2.92 | 26.80 | 99.40 | 160.34 | 237.16 | 235.61 | 0.88 | 4.58 | -52.75 | -8.02 | 11.40 |
| RW20 | 11.6 | 8.57 | 926.40 | 105.40 | 4.20 | 86.20 | 71.50 | 136.50 | 233.60 | 393.80 | 1.90 | 4.90 | -52.92 | -7.62 | 8.04 |
| RW21 | 11.0 | 8.35 | 1577.00 | 163.20 | 4.10 | 98.70 | 175.10 | 327.40 | 107.70 | 708.60 | 1.60 | 4.40 | -53.04 | -7.28 | 5.20 |
| RW22 | 11.8 | 8.44 | 1410.00 | 155.70 | 3.95 | 98.50 | 185.60 | 405.10 | 244.00 | 435.30 | 1.22 | 4.58 | -51.90 | -7.62 | 9.06 |
| RW23 | 12.4 | 8.94 | 1672.00 | 158.80 | 5.21 | 37.30 | 372.70 | 631.30 | 173.00 | 452.60 | 1.10 | 4.92 | -32.80 | -3.40 | -5.60 |
| GW01* | 16.9 | 7.92 | 704.80 | 158.95 | 3.16 | 30.50 | 32.62 | 26.30 | 310.23 | 308.90 | 2.90 | 4.11 | -67.57 | -10.08 | 13.07 |
| GW02* | 5.2 | 7.95 | 265.00 | 34.80 | 2.49 | 25.01 | 26.90 | 22.70 | 225.77 | 43.29 | 0.64 | 3.92 | -84.70 | -12.46 | 14.98 |
| GW03 | 7.5 | 7.10 | 742.00 | 91.73 | 4.01 | 87.35 | 95.77 | 60.56 | 787.16 | 62.40 | 0.50 | 4.15 | -79.30 | -11.71 | 14.38 |
| GW04 | 10.5 | 7.32 | 452.00 | 47.65 | 3.68 | 38.39 | 58.66 | 54.93 | 328.49 | 71.06 | 1.26 | 4.13 | -62.30 | -8.61 | 6.58 |
| GW05 | 10.8 | 7.23 | 400.00 | 42.22 | 3.12 | 39.18 | 54.24 | 63.91 | 264.08 | 64.05 | 1.31 | 4.00 | -64.30 | -9.83 | 14.34 |
| GW06 | 11.8 | 8.06 | 1160.00 | 94.90 | 1.80 | 94.90 | 166.00 | 247.13 | 190.38 | 508.02 | 4.69 | 4.20 | -53.73 | -7.75 | 8.29 |
| GW07 | 12.5 | 7.79 | 1382.00 | 107.50 | 2.20 | 122.00 | 199.00 | 320.87 | 161.09 | 681.05 | 6.04 | 4.40 | -53.82 | -7.95 | 9.74 |
| GW08 | 12.20 | 7.81 | 967.00 | 88.30 | 1.50 | 62.90 | 87.90 | 144.41 | 214.79 | 257.46 | 2.42 | 3.90 | -55.12 | -8.27 | 11.08 |

*For groundwater samples, an asterisk (\*) represents spring water, while the others are taken from wells.*

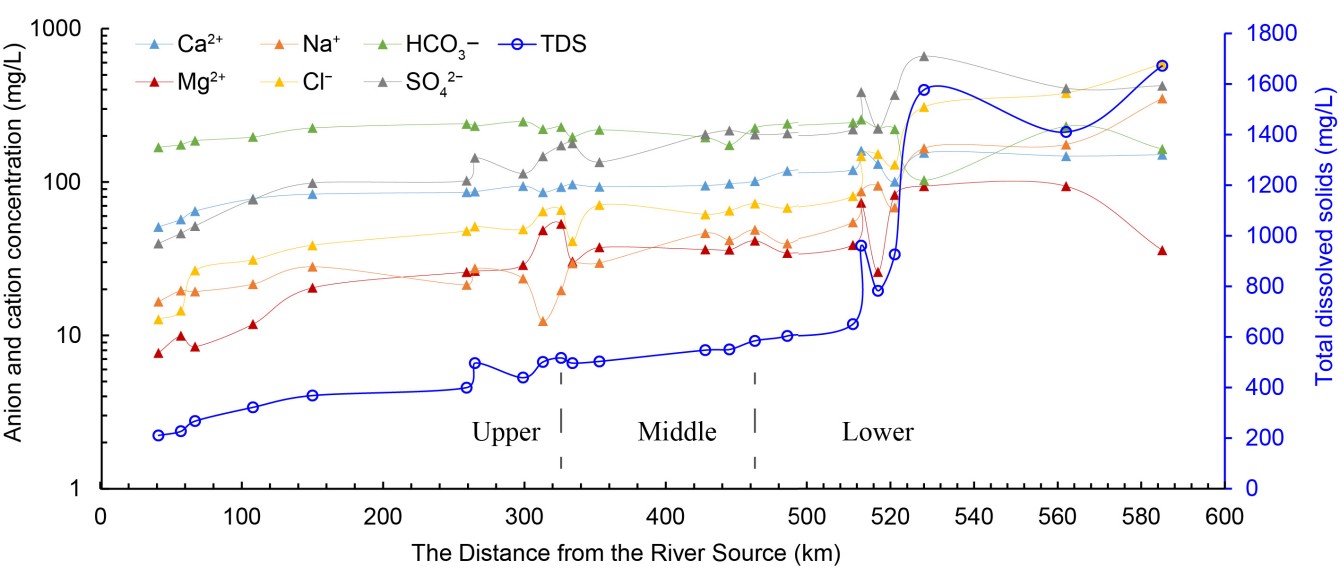

**Figure 4: Hydrochemical characteristics changes along Shule River. The x-axis does not use a uniform scale. The left side is the upstream area (samples 1 to 10), the middle is the midstream area (samples 11 to 15), and the right side is the downstream area (samples 16 to 23).**

In the upstream river water within the study area, the average mass concentration of major cations follows the order of $Ca^{2+}$,

$Mg^{2+}$, $Na^+$, and $K^+$, from highest to lowest. Conversely, in the middle and downstream areas, the average mass concentration of major cations exhibits a different sequence, with $Ca^{2+}$, $Na^+$, $Mg^{2+}$, and $K^+$. Concerning anions, in both the upstream and

middle areas, the predominant ion mass concentration is arranged in the order of $HCO_3^-$, $SO_4^{2-}$, and $Cl^-$. However, in the downstream area, the sequence changes to $SO_4^{2-}$, $Cl^-$, and $HCO_3^-$. As a result, the hydrochemical composition of the river water in the upstream area is categorized as $Ca^{2+}$- $HCO_3^-$ type, while in the middle area, it falls into the $Ca^{2+}$- $Mg^{2+}$- $HCO_3^-$-$SO_4^{2-}$ category. In the downstream region, the river water assumes a $Ca^{2+}$- $Na^+$- $SO_4^{2-}$-Cl type (Fig. 5).

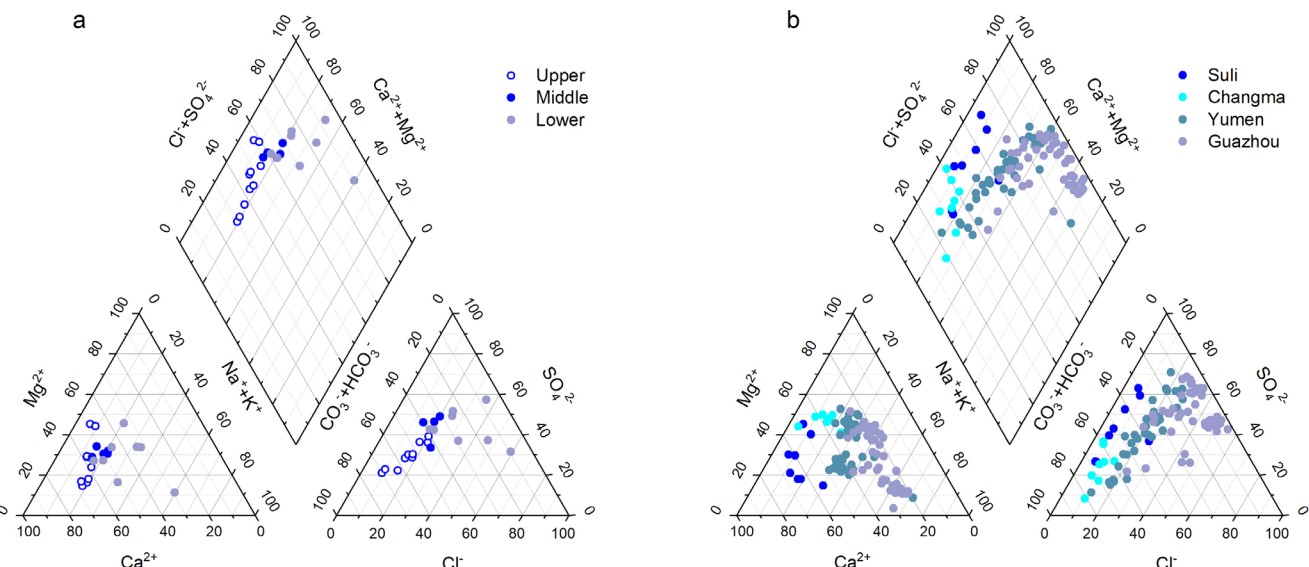

**Figure 5: Piper trilinear plots for the chemical compositions of the river water and groundwater in the study area. In order to more clearly understand the changes in the hydrochemical characteristics of groundwater, groundwater samples were classified according to the sampling locations. The upper reaches include Suli and Changma, the middle reaches include Yumen, and the lower reaches include Guazhou.**

The Gibbs diagram depicted in Fig. 6 is constructed based on the correlation between the primary ion ratios and TDS in the Shule River water. All river water samples fall within the Gibbs model, indicating that the solutes in the Shule River water are largely impacted by rock weathering and evaporation. Furthermore, the TDS values of river water mainly range from 100 to 1000 mg/L, and the mean $Na^+/(Na^+ + Ca^{2+})$ ratio is 0.19 in the upstream river water, 0.26 in the middle stream river water, and 0.41 in the downstream river water. Additionally, the mean $Cl^-/(Cl^- + HCO_3^-)$ ratio is 0.23 in the upstream river water, 0.34 in the middle stream river water, and 0.58 in the downstream river water. This indicates that the primary sources of ions in river water are predominantly controlled by rock weathering processes.

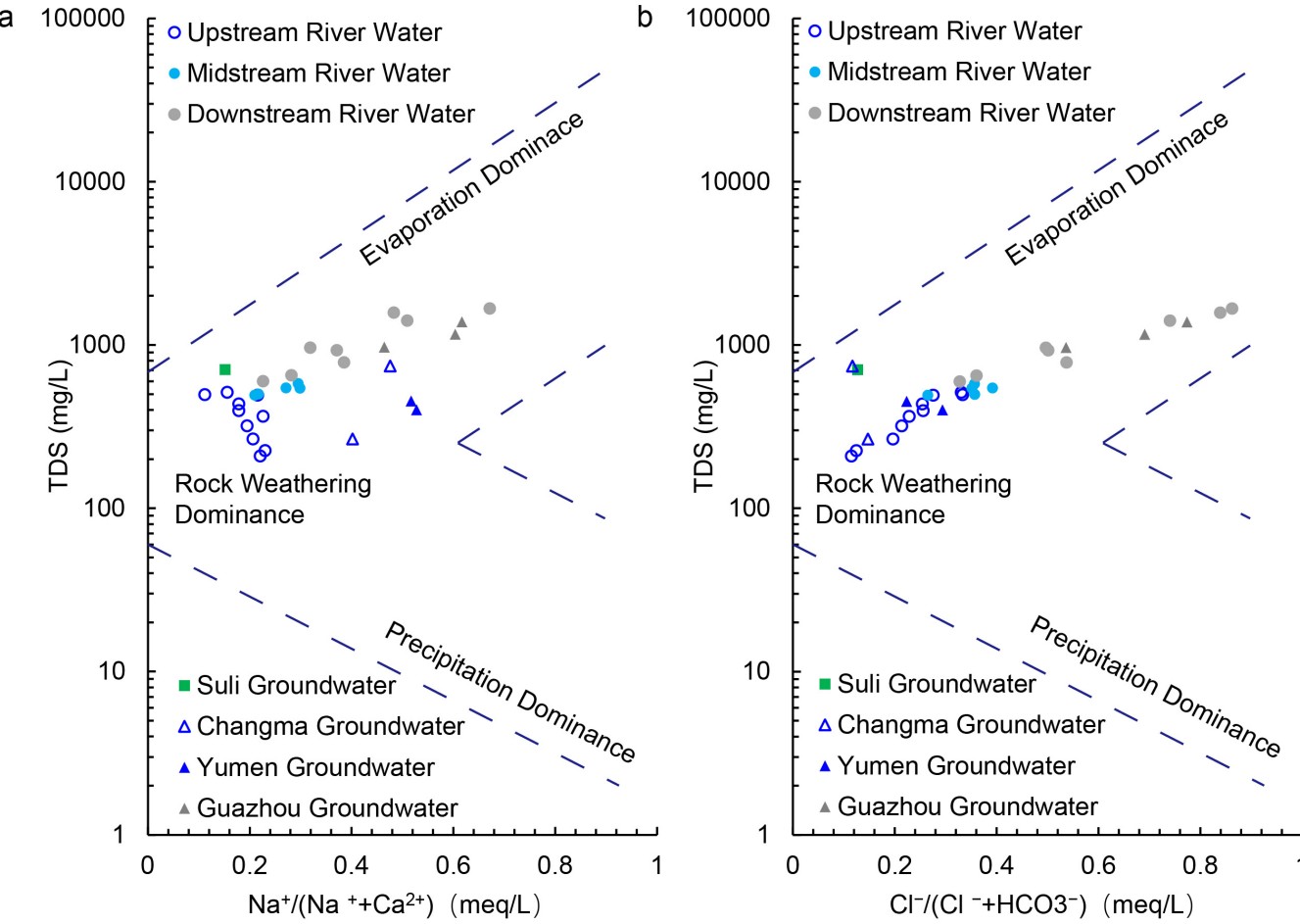

**Figure 6: Gibbs diagrams generated based on Shule River water data and groundwater data, (a) TDS versus N a⁺, (b) TDS versus Cl⁻.**

To clarify further into understanding the potential impact of various mineral weathering processes on river water ions within the study area, a sodium-normalized mixing diagram was made in Fig. 7. The results reveal that river water samples predominantly cluster between silicate and carbonate rocks, with a closer alignment to silicate rock types. This suggests that the ions in river water primarily originate from the weathering of silicate rocks, but the contribution from the weathering products of carbonate rocks should not be underestimated. It's noteworthy that the $Ca^{2+}/Na^+$, $Mg^{2+}/Na^+$, and $HCO_3^-/Na^+$ ratios in the upstream river water of the study area exceed those in the middle and downstream river water. This signifies that carbonate rock weathering in the upstream region exerts a more pronounced impact on the major ions in the river water. This phenomenon is not only linked to the extensive exposure of carbonate rock formations in the upstream region but also to the fact that, under similar natural conditions, carbonates exhibit significantly higher solubilities (12–40 times) than silicates, making them more susceptible to weathering (Meybeck, 1987).

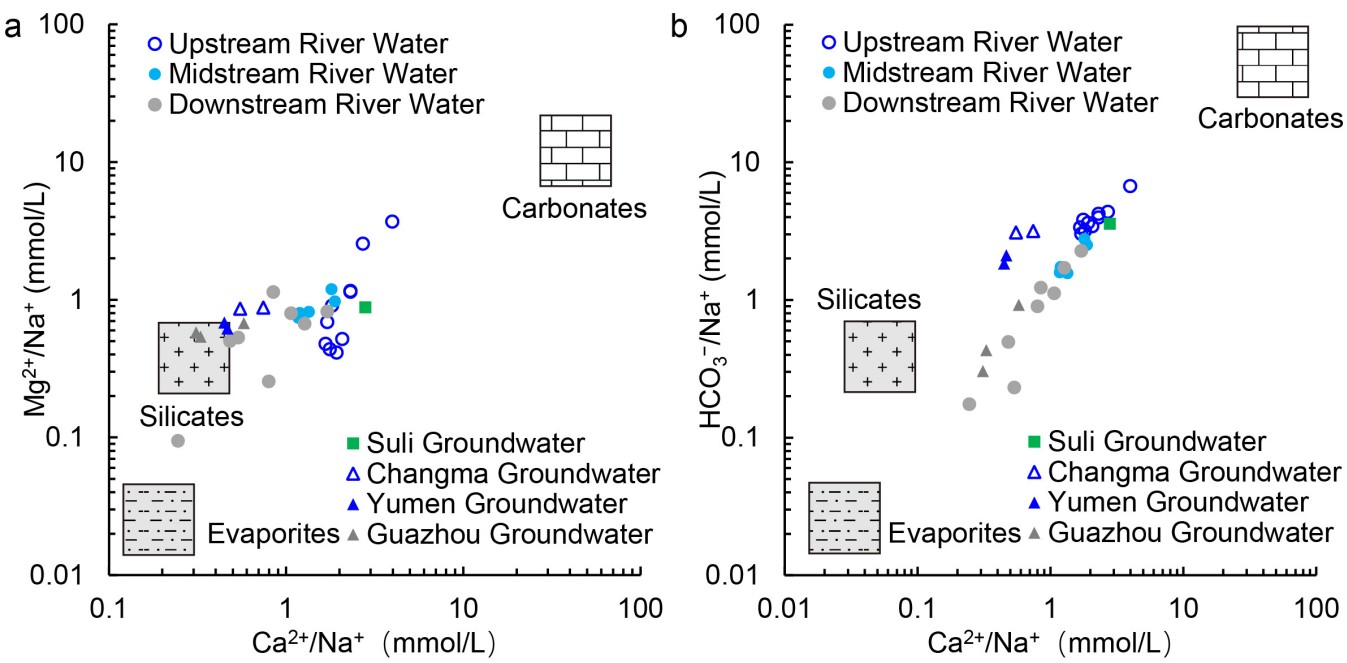

**Figure 7: Mixing diagram of the Na-normalized molar ratios of (a) Ca$^{2+}$ versus Mg$^{2+}$ and (b) Ca$^{2+}$ versus HCO$_3^-$ in the Shule river. The data for the three endmembers, i.e., carbonates, silicates and evaporites, are obtained from Gaillardet et al. (1999).**

### 3.2 Correlation coefficients among major ions in river water

We have calculated the correlation coefficients among ions in the Shule River water and presented them in Table 2. Values closer to 1 in this table indicate a stronger positive correlation between the respective ion pairs.

**Table 2 Correlation matrix of hydrochemical compositions in Shule River water**

|  | pH | TDS | Ca$^{2+}$ | K$^+$ | Mg$^{2+}$ | Na$^+$ | Cl$^-$ | HCO$_3^-$ | SO$_4^{2-}$ | NO$_3^-$ | SiO$_2$ |
|---|---|---|---|---|---|---|---|---|---|---|---|
| pH | 1 | | | | | | | | | | |
| TDS | 0.415* | 1 | | | | | | | | | |
| Ca$^{2+}$ | 0.240 | 0.892** | 1 | | | | | | | | |
| K$^+$ | 0.355 | 0.837** | 0.751** | 1 | | | | | | | |
| Mg$^{2+}$ | 0.053 | 0.758** | 0.731** | 0.621** | 1 | | | | | | |
| Na$^+$ | 0.570** | 0.907** | 0.745** | 0.780** | 0.439* | 1 | | | | | |
| Cl$^-$ | 0.530** | 0.935** | 0.765** | 0.767** | 0.531** | 0.987** | 1 | | | | |
| HCO$_3^-$ | -0.170 | -0.169 | 0.126 | -0.162 | 0.074 | -0.248 | -0.208 | 1 | | | |
| SO$_4^{2-}$ | 0.276 | 0.942** | 0.878** | 0.809** | 0.859** | 0.741** | 0.770** | -0.214 | 1 | | |
| NO$_3^-$ | -0.354 | -0.578** | -0.523* | -0.490* | -0.423* | -0.575** | -0.571** | 0.071 | -0.532** | 1 | |
| SiO$_2$ | 0.072 | 0.282 | 0.127 | 0.395 | 0.129 | 0.329 | 0.309 | -0.090 | 0.215 | 0.079 | 1 |

Significant value (** $p < 0.01$, * $p < 0.05$)

It is evident that the correlation coefficients between $Ca^{2+}$, $K^+$, $Na^+$, $Cl^-$, $SO_4^{2-}$ and TDS in the river water are highly significant, all exceeding 0.8. This suggests their substantial contribution to the salinity of the river water. Furthermore, the correlation coefficients between $Ca^{2+}$, $Mg^{2+}$ and $SO_4^{2-}$ surpass 0.85, which indicates that these three ions may originate from the dissolution of $CaSO_4$ and $MgSO_4$. Notably, the correlation coefficient between $Na^+$ and $Cl^-$ is 0.987, very close to 1, and their respective correlation coefficients with TDS are also greater than 0.9. This implies that $Na^+$ and $Cl^-$ share a common source and make a pronounced contribution to the salinity of the river water. It is likely that the $Na^+$ and $Cl^-$ sources may mainly be the inputs of the sea salt brought by the atmospheric circulation and the salt particles of the air in the study area. In addition, the correlation coefficients between $NO_3^-$ and the other ions are all less than 0.4, indicating a potentially closer association with human activities rather than natural process (Huang et al., 2014).

### 3.3 Hydrochemical characteristics of groundwater

Considering the differences in sampling locations and previous studies (See Table S1), groundwater samples from the SRB have been grouped into four categories: Suli and Changma groundwater from the upstream Qilian Mountains, Yumen groundwater from the midstream Hexi Corridor, and Guazhou groundwater from the downstream plain (Fig. 5). Suli groundwater (n = 12, including samples reported by Xie et al. (2024)) exhibits a pH range of 7.10 to 8.20 (mean 7.87) and TDS between 304 and 977.9 mg/L. The dominant controlling cation is $Ca^{2+}$, while $HCO_3^-$ is the principal anion (Fig. S1). Changma groundwater (n = 8, incorporating data from Wang et al. (2015) and Xie et al. (2024) ) shows hydrochemical characteristics closely resembling those of the Suli samples. Measured pH values average 7.81, with TDS spanning 235 to 742 mg/L. The chemistry type is $Ca^{2+}$- $HCO_3^-$ type (Fig. S2). Yumen goundwater (n = 37, including He et al. (2015); Wang et al. (2015); Xie et al. (2024)) displays a broader pH distribution from 6.84 to 8.29 (mean 7.67) and TDS values ranging from 255 to 1117 mg/L (mean 506.85 mg/L). The hydrochemical content was not dominated by speci cations, whereas $HCO_3^-$ and $SO_4^{2-}$ together constitute the main anionic species (Fig. S3). In the downstream Guazhou plain (n = 50, comprising He et al. (2015); Wang et al. (2016); Xie et al. (2024)), pH varies between 6.92 and 8.10 (mean 7.69) and TDS ranges widely from 449 to 7142.4 mg/L (mean 1499.65 mg/L). These groundwater samples are of $Na^+$- $SO_4^{2-}$ type (Fig. S4). Overall, groundwater in the SRB is a weak alkaline nature. Notably, TDS levels are lowest in spring water from the Qilian Mountains region and highest in groundwater from the downstream Guazhou plain. Groundwater chemistry types vary, with the Qilian Mountains region classified as $Ca^{2+}$- $HCO_3^-$ type, and Guazhou as $Na^+$- $SO_4^{2-}$ type (Fig. 5).

### 3.4 Stable isotopic composition of river water and groundwater

According to the Table 1, in the upper reaches of the Shule River, the $\delta D$ values of river water range from −78.02‰ to −58.25‰, with an average value of −62.21‰, while the $\delta^{18}O$ values range from −12.12‰ to −8.70‰, averaging at −9.54‰. The d-excess falls within the range of 8.7‰ to 18.96‰, with an average of 14.10‰. To the middle reaches of the river, the average $\delta D$ and $\delta^{18}O$ values are −56.22‰ and −8.54‰, respectively, with a d-excess average of 12.11‰. In the downstream area, the average $\delta D$ and $\delta^{18}O$ isotope values in river water are −50.51‰ and −7.05‰, respectively, and the d-excess averages at 5.87‰.

Concerning groundwater, in the Qilian Mountains region, the $\delta$D values range from −84.70‰ to −67.57‰, with an average of −77.19‰, while the $\delta^{18}$O values range from −12.46‰ to −10.08‰, averaging at −11.42‰. The d-excess falls within the range of 8.7‰ to 18.96‰, with an average of 10.50‰. In the Yumen area of the Hexi Corridor, the average $\delta$D and $\delta^{18}$O isotope values of groundwater are −63.30‰ and −9.22‰, respectively, with a d-excess average of 12.11‰. Conversely, in the

305 Guazhou area, the average $\delta$D and $\delta^{18}$O isotope values of groundwater are −54.23‰ and −7.99‰, respectively, with a d-excess average of 9.7‰.

## 4 Discussion

### 4.1 Characteristics and hydrological implications of river water stable isotopes

#### 4.1.1 Identification of river water recharge sources

Precipitation is not only a key component of the hydrological cycle but also the principal source of recharge for both rivers and groundwater (Galewsky et al., 2016; Wang et al., 2022). In isotope-based hydrological studies, the $\delta$D and $\delta^{18}$O values of river water and groundwater are routinely compared with those of precipitation to clarify their mutual relationship (Wang et al., 2024a). Because these isotopic signatures are strongly modulated by climatic and physiographic gradients within the SRB, a single Local Meteoric Water Lines (LMWL) cannot represent the entire basin. Therefore, separate LMWLs must be

established via linear regression of $\delta$D against $\delta^{18}$O in precipitation for both the high-elevation Qilian Mountains headwaters and the low-elevation, arid Hexi Corridor.

In the upper reaches of the SRB (from the headwaters to Changma Reservoir), Guo et al. (2022) established a LMWL-U based on monthly amount-weighted isotope measurements of precipitation at two sampling sites, yielding the relationship: $\delta$D = 7.40 × $\delta^{18}$O + 9.86 ($R^2$ = 0.97, n = 66). Although the slope of this LMWL is slightly lower than that of the Global Meteoric Water

Line (GMWL: $\delta$D = 8 × $\delta^{18}$O + 10, (Craig, 1961)), the intercept of 9.86‰ is statistically indistinguishable from the GMWL intercept. In contrast, long-term isotope observations are lacking for the middle and lower reaches the middle and lower reaches of the SRB. To represent the regional meteoric water line in these downstream areas, we adopt the IAEA–GNIP-derived relation for Zhangye, namely LWML-ZY: $\delta$D = 7.50 × $\delta^{18}$O + 2.70, (IAEA and WMO, 2006). Zhangye city lies approximately 350 kilometers east of the study area, like the middle and lower Shule River reaches, is situated within the Hexi Corridor; its

prevailing climatic and geographic characteristics closely mirror those of the downstream basin. Consequently, the hydrogen–oxygen isotope composition of precipitation in Zhangye can be taken as representative of the Hexi Corridor's meteoric water signature under hyperarid conditions, where the lower slope and reduced intercept underscore enhanced evaporative enrichment and continental moisture recycling.

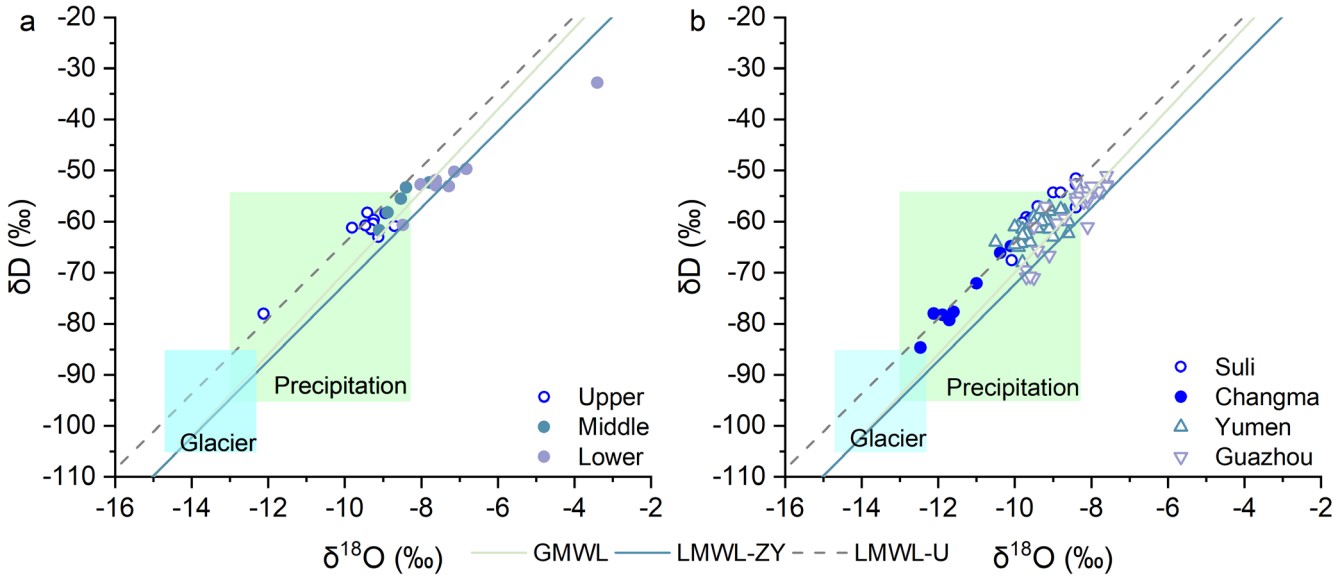

**Figure 8: δD and δ18O relationships of river water (a) and groundwater (b) in the Shule River Basin, including previously published groundwater data (Xie et al., 2022; Wang et al., 2015; He et al., 2015), plotted against the Global Meteoric Water Line (GMWL), the local meteoric water line in the upper reaches (LMWL-U), and that in the middle and lower reaches (LMWL-ZY). The pale green rectangle denotes the δD and δ18O range of precipitation in the upper basin, while the pale blue rectangle denotes the δD and δ18O range of glacier meltwater (Guo et al., 2022).**

The isotopic analysis results were plotted together with the GMWL and the corresponding LMWLs in Fig. 8. River water samples collected from the upper reaches of the Qilian Mountains plot tightly along the LMWL-U, suggesting that meteoric water in the Qilian Mountains is the major source of the river water (Fig. 8a). Nevertheless, the pronounced seasonal variability of precipitation in the headwaters means that direct rainfall cannot sustain baseflow during the dry season (October–April). In this context, glacial meltwater and groundwater become additional contributors. Their nearly overlapping isotopic signatures

(Fig. 8) highlight a close hydrological connection. Using combined δ18O, d-excess and chloride tracers, Guo et al. (2022) estimated that groundwater a provides 45–100% of the monthly discharge over a hydrological year, while glacial meltwater and direct precipitation contribute 2–23% and 2–32%, respectively. These results underscore the pivotal role of groundwater in sustaining upstream flow. Isotopic signatures of river water in the middle reaches largely mirror those of the upstream reach: most samples still plot along the LMWL-U (Fig. 8), but they are displaced slightly to the right, indicating heavier values. This

enrichment reflects two concurrent influences. Progressive evaporation during downstream transport fractionates the remaining flow toward heavier isotopes, and abundant spring discharge emerging along the margin of the Yumen alluvial fan mixes with the channel water, superimposing its own isotopic imprint and further shifting the mid-reach distribution. Interestingly, the downstream river water isotopes fall apart from the LMWL-U into the LMWL-ZY and these isotopic values are more positive. Theoretically, the δD and δ18O values in river water in the upstream area of the study region would purely

reflect the isotopic characteristics of precipitation in the Qilian Mountains. In contrast, in the downstream areas, these would manifest as a mixture of precipitation from the Qilian Mountains and the Hexi Corridor plains. This explains why the isotopic

scatter plot of river water gradually shifts from the LMWL-U to the LMWL-ZY. Additionally, Fig. 8 illustrates that the isotopic composition of river water in the study area exhibits a progressively positive trend from upstream to downstream. This trend is likely due to evaporation effects experienced by the river water after a prolonged runoff process. Kalvāns et al. (2020) found that evaporation of water impounded on the soil surface is an important mechanism leading to isotopic enrichment of surface water. Considering the scarce rainfall in the Hexi Corridor region and the relatively weak contribution of precipitation to river runoff, we believe t the strong evaporation in the study area, whether occurring in the river water or in the soil water, is the main factor affecting the isotopic changes in the river water along the runoff process. In other words, due to the intense evaporation in the SRB, the isotopic values of river water become increasingly positive as it flows from the upstream to the downstream, progressively diverging from the LMWL-U.

In summary, the Shule River's runoff is generated chiefly from the Qilian Mountains in the upper reaches, corroborating previous studies (Wu et al., 2021; Zhou et al., 2021). This source pattern typifies mountain–basin landscapes in arid regions worldwide, where both surface water and groundwater (Wang et al., 2015) are produced and sustained in the high altitude mountainous areas due to comparatively abundant precipitation, snowmelt and glacial meltwater.

### 4.1.2 Spatial variation in river water isotopic characteristics

Using the linear regression method, we found that the $\delta^{18}$O-altitude gradient for the Shule River water is approximately −0.08‰/100m, suggesting that the $\delta^{18}$O values in river water will decrease with increasing elevation. Such negative slopes are characteristic of the meteoric water altitude effect (Dansgaard, 1964) and are typically produced by Rayleigh fractionation during orographic condensation (Li and Garzione, 2017; Bershaw et al., 2012). Nevertheless, the $\delta^{18}$O-altitude gradient in the SRB is markedly weaker than those reported for adjacent basin. In comparison, in the Heihe River basin, which is located about 300km southeast of the study area, the altitudinal isotopic gradient of precipitation is −0.18‰/100m (Wang et al., 2009). Meanwhile, the altitudinal isotopic gradient of precipitation and river water ranges from −0.1‰/100m to −0.3‰/100m in the Qinghai-Tibet Plateau region (Li and Garzione, 2017). Although there is still a decreasing trend in the $\delta^{18}$O values of river waters with increasing elevation, it is greatly attenuated from the typical −0.3‰/100m to −0.08‰/100m in the study area, indicating that other fractionation mechanisms, such as addition of recycled surface water and sub-cloud evaporation, significantly affect the isotopic composition of the river water in the SRB.

In addition, the d-excess values of Shule River water decrease systematically from the headwaters to the lower reaches. Because d-excess quantifies the relationship between $\delta$D and $\delta^{18}$O, it records both source-air humidity and post-depositional evaporative fractionation and reflects certain characteristics of the hydrological environment. Under arid or semi-arid conditions, prolonged surface residence enhances evaporation of river water, enriching the remaining water in heavy isotopes and driving d-excess to lower values. Although the basin scale mean d-excess of the Shule River (10.8 ‰) approximates the global precipitation average (~10 ‰), downstream samples commonly fall below 10 ‰ and may even become negative, clear evidence of evaporative enrichment. These observations demonstrate that progressive evaporation exerts a persistent and increasingly dominant influence on the isotopic composition of Shule River water.

## 4.2 Factors controlling river water hydrochemistry

The potential sources of major ions in the river water include atmospheric transport of sea salt components, weathering products of soluble rocks (comprising evaporites, silicates, carbonates, and sulfides), and pollutants generated by human activities. In the upper reaches of the Shule River, human presence is scarce, while human activities are more frequent in the middle and lower reaches. Consequently, the major ions in the river water may be influenced by both natural processes and human activities. The Gibbs diagrams has proved that the sources of ionic components in river water are maily from the weathering of carbonate rocks and silicate rocks (Fig. 7).. In-depth analysis of the correlation between multi-ions can further understand the causes of river water hydrochemistry and the control mechanisms of its evolution.

To gain insight into the interrelationships among ions in Shule River water, $Na^+$ versus $Cl^-$ and ($Ca^{2+}$ $Mg^{2+}$) versus ($HCO_3^-$+ $SO_4^{2-}$) diagrams were created and presented in Fig. 9. It can be seen that the $Na^+/Cl^-$ points in upstream river water deviates from the 1:1 line, while points for downstream river water samples closely align with the 1:1 line. Factors affecting $Na^+$ in water and their ratio to $Cl^-$ include evaporite dissolution, ion-exchange adsorption, and human activities. Whereas human activities are very rare in the very upstream area of the Shule River, we consider that in the very upstream region, the dissolution of silicates predominantly influences the source of Na, while downstream, the dissolution of salt rocks becomes the dominant factor. As a result, the $Na^+$ versus $Cl^-$ ratios show a clear difference between upstream and downstream (Fig. 9a). The average $Na^+/Cl^-$ ratio in river water is 1.16, which is similar to the world average ratio in seawater (1.15). This shows that the sea salt carried by the atmospheric circulation has a certain influence on the ion composition of Shule River water. Furthermore, the concentration of $Na^+$ exceeded the concentration $Cl^-$, which also may indicate that halite dissolution was not the only source for $Na^+$. For example, in the case of groundwater recharging a river, the concentration of $Na^+$ in the water will increase after cation exchange occurs after the pore water flows through the formation. Moreover, silicate weathering could release $Na^+$ into the river water (Guo et al., 2015). In addition, a linear relationship is found between ($Ca^{2+}$ $Mg^{2+}$) and ($HCO_3^-$+ $SO_4^{2-}$), with their ratios clustering in the range of 0.88 to 1.24, and an average value of 1.06. If the ratio of ($Ca^{2+}$ $Mg^{2+}$) and ($HCO_3^-$+ $SO_4^{2-}$) is strictly 1:1, then it can be assumed that these four ions in river water are controlled by the dissolution of carbonate rocks. However, it is clear that some points deviate from the 1:1 line as shown in Fig 9b. Thus, in addition to carbonate rocks dissolution, silicate weathering plays a role in influencing the levels of these four ions in river water, as previously mentioned.

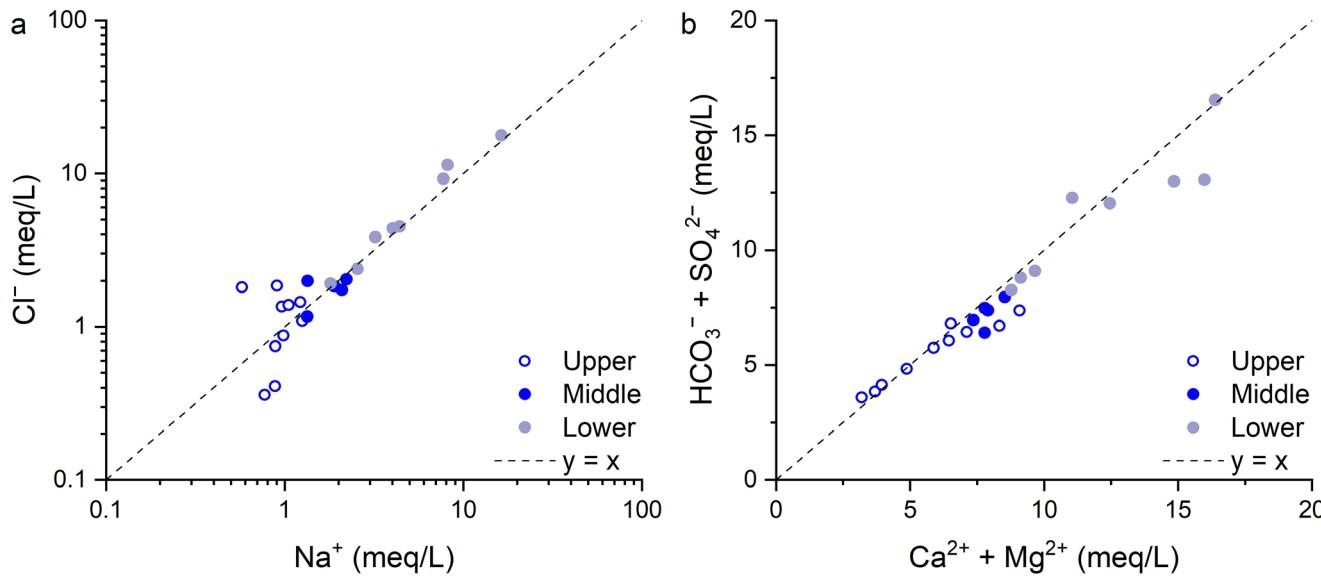

Figure 9: Bivariate ionic correlations in Shule River water: (a) Na$^+$ versus Cl$^-$ concentrations, (b) (Ca$^{2+}$ Mg$^{2+}$) versus (HCO$_3^-$+ SO$_4^{2-}$) concentrations.

## 4.3 The Groundwater Recharge Sources and Evolutionary Processes

As the SRB is situated in an arid region with scarce rainfall, its water resources predominantly comprise river water and groundwater. Groundwater is known for its superior quality, extensive spatial distribution, and comparatively stable hydrodynamic characteristics within aquifers. Given the limited number of groundwater samples collected in this study, we propose to integrate existing data from thd study area for a more robust analysis, aiming to systematically elucidate regional groundwater cycle processes.

### 4.3.1 Groundwater Recharge Sources

The groundwater within the SRB was broadly divided into segments corresponding to the upper, middle, and lower reaches of the river: Suli groundwater and Changma groundwater (upper reaches), Yumen groundwater (middle reaches), and Guazhou groundwater (lower reaches). By integrating the groundwater isotope data obtained in this study with previously published datasets (Table S1), we find that samples from the upper reaches of the Shule River plots tightly along the LMWL-U (Fig. 8). This alignment indicates a direct genetic link between precipitation and local groundwater in the Qilian Mountains. Moreover, the $\delta$D and $\delta^{18}$O values of groundwater in the upper reaches largely fall within the isotopic rectangular range characterizing Qilian Mountains' precipitation (Fig. 8b), confirming precipitation as a major recharge source. However, a comparison of isotopic compositions from Suli and Changma aquifers reveals an unexpected pattern: groundwater from Suli exhibits consistently less-negative $\delta$D and $\delta^{18}$O values than Changma water, despite Suli's proximity to the headwaters and Changma's

downstream position. This observation contradicts the expectation that downstream groundwater should simply inherit the upstream isotopic signature.

The apparent paradox can be explained by contrasting hydrogeological settings. Suli groundwater resides in alluvial deposits within the mountain valleys of the Shule River, maintaining a strong hydraulic connection with both local precipitation and streamflow. In contrast, the Changma aquifer lies within an alluvial-proluvial fan that is recharged predominantly by rapid

infiltration of glacier meltwater. High hydraulic permeability of moraine and till deposits, coupled with low evaporation at higher elevations, allows meltwater to percolate quickly into the Changma aquifer. The strongly depleted isotopic signature of glacier meltwater therefore dominates the isotopic composition of Chama groundwater, rendering it isotopically lighter than the Suli supply. Moreover, the two aquifer units are hydraulically disconnected, precluding any direct transfer or inheritance of isotopic composition from Suli to Changma.

Most groundwater samples from the middle reaches of the SRB, near Yumen City, plot along the LMWL-U. Compared to groundwater in the upstream Changma alluvial fan, the Yumen groundwater exhibits more enriched isotopic values (Fig. 8b), yet both align with the same meteoric line, suggesting a downstream isotopic inheritance. This pattern may indicate a potential hydraulic connection between the two groundwater systems. However, analysis of the hydrogeological setting reveals the presence of bedrock ridges and fault zones that likely inhibit direct groundwater flow between the aquifers. In contrast, the

isotopic values of Yumen groundwater and upstream river water show substantial overlap (Fig. 8), supporting the interpretation that, after emerging from the mountains at Changma, river water effectively recharges the local aquifer due to the high permeability of gravel deposits in the riverbed and in the upper part of the Yumen alluvial fan. Furthermore, groundwater isotopic values plot to the left of the LMWL ZY, indicating that local precipitation contributes only marginally to groundwater recharge. This interpretation is consistent with previous studies. Wang et al. (2015) also pointed out that in the middle reaches

of the Shule River, the recharge effect of local atmospheric precipitation on groundwater is very weak or even negligible. Guo et al. (2015) concluded that the contribution of local atmospheric precipitation to groundwater is only about 1%.

As we move downstream along the Shule River, the $\delta$D and $\delta^{18}$O composition of groundwater gradually diverges from the LMWL-U. However, a significant portion still resides on the left side of the LMWL-ZY. In the downstream region, the isotope composition of groundwater exhibits substantial overlap with surface water, indicating a robust hydraulic linkage between the

455 two. Considering the hydrogeological conditions and river characteristics here, and combining the previous research findings of Wang et al. (2016) who use used hydrochemical evolution to study the groundwater in Guazhou, it can be reasonably concluded that groundwater in the downstream region is primarily recharged by Shule River water, with contributions from local precipitation in the Hexi Corridor recharge being negligible.

### 4.3.2 The Hydrochemical Evolution of Groundwater

The hydrochemical characteristics of groundwater in the study area are influenced not only by their recharge sources but, to a larger extent, by the interactions between groundwater and the aquifer during the groundwater flow. Based on the hydrochemical data obtained in this study, integrated with previously published datasets (Table S1), the dominant geochemical

processes governing groundwater evolution were identified through a comprehensive ion correlation analysis. Consistent with patterns observed in river water, a good linear relationship was found between $Na^+$ and $Cl^-$ concentrations, closely aligned

along the 1:1 stoichiometric line (Fig. 10a), suggesting that halite dissolution is the primary source of these ions. In addition, a significant linear correlation was observed between ($Ca^{2+}$+ $Mg^{2+}$) and ($HCO_3^-$+ $SO_4^{2-}$) ($R^2$ = 0.9125; Fig. 10b), indicating that the dissolution of carbonate and sulfate minerals, particularly dolomite, calcite, and gypsum, is the principal contributor to the presence of these ions in groundwater.

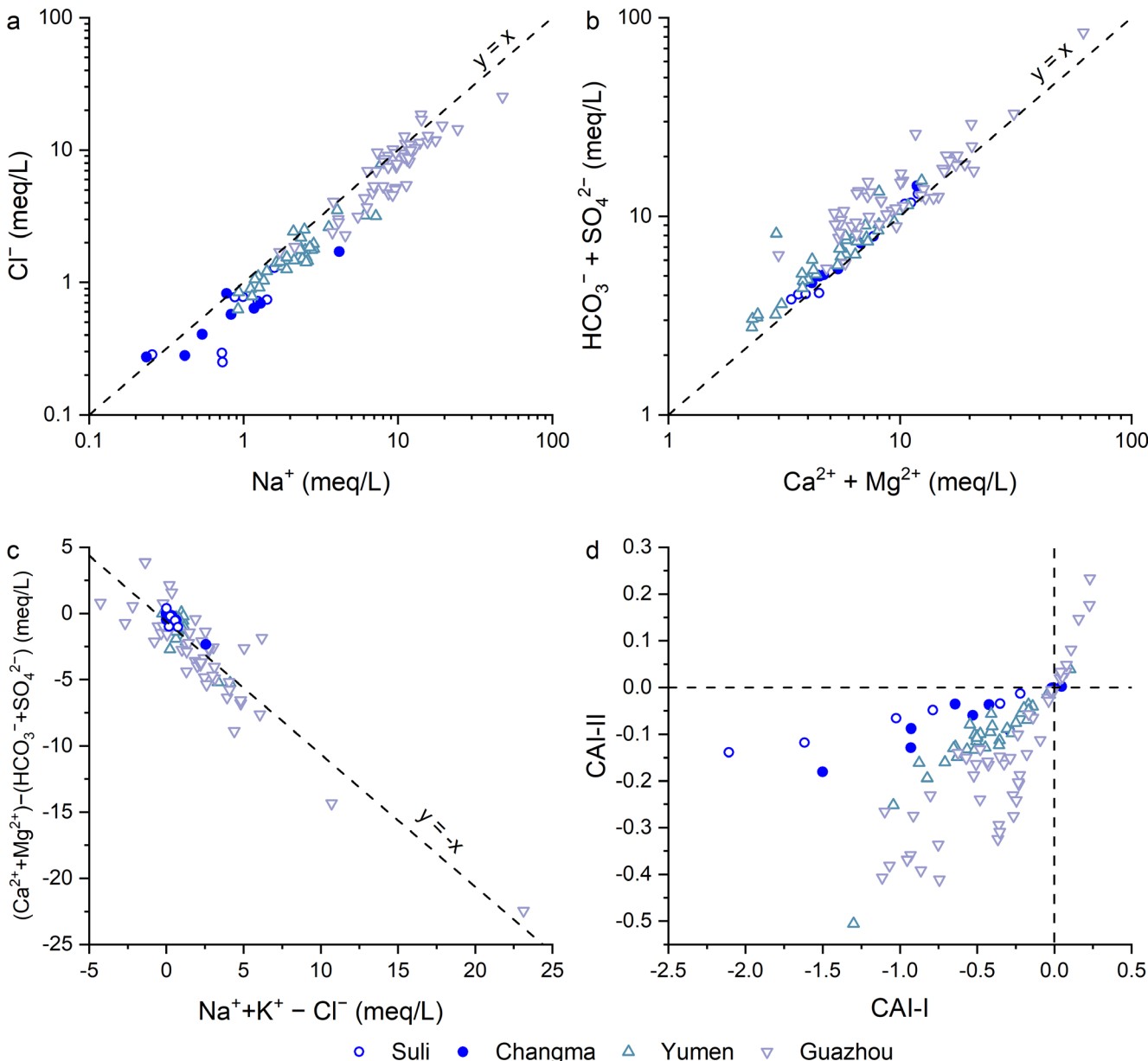

**Figure 10: Bivariate ionic correlations and Chloro-Alkaline index of groundwater in the Shule River Basin.**

However, certain groundwater samples deviate from the ideal 1:1 stoichiometric lines in both Figures 9a and 9b. Previous studies (Guo et al., 2015; Wang et al., 2015; Wang et al., 2016) attributed these deviations to the dissolution of mirabilite, which introduces additional $Na^+$ and $SO_4^{2-}$ into the system. Further evidence for geochemical processes beyond simple mineral dissolution is provided by the relationship between ionic differentials, specifically $(Ca^{2+} + Mg^{2+}) - (HCO_3^- + SO_4^{2-})$ plotted against $(Na^+ + K^+) - Cl^-$. The samples align along a line with a slope of approximately $-1$ (Fig. 10c), which is indicative of active cation exchange reactions. This inference is corroborated by the Chloro-Alkaline Indices (CAI 1 and CAI 2) defined by Schoeller (1965). As shown in Fig. 10d, the majority of groundwater samples exhibit negative CAI values, suggesting that $Na^+$ and $K^+$ from the aquifer matrix are replacing $Ca^{2+}$ and $Mg^{2+}$ in groundwater. These results collectively indicate that in addition to mineral dissolution, cation exchange reactions also play a non-negligible role in the geochemical evolution of groundwater in the study area.

## 4.4 Water quality for drinking and irrigation

The SRB, characterized by a significantly higher annual evaporation rate compared to precipitation, stands as one of the most arid regions in China. Nevertheless, this area accommodates an extensive irrigation network spanning over 1.3 million acres, providing a robust foundation for local socioeconomic development and human livelihood (Ma et al., 2018). Consequently, the water quality of both surface water and groundwater in the SRB directly impacts regional ecological security and sustainable socioeconomic progress. In this context, we evaluate the suitability of river water and groundwater for drinking and irrigation based on parameters such as TH, $Na^+\%$, SAR, and $NO_3^-$ concentration.

In the Shule River's headwaters region, 50% of the water samples are classified as slightly hard, and the remaining 50% are categorized as hard water. As we move to the middle reaches, the water is consistently characterized as hard, and in the downstream areas, it becomes very hard, demonstrating a gradual deterioration in water quality from upstream to downstream. A similar pattern is observed in the groundwater quality within the study area. However, it's worth noting that in the arid SRB, the water quality remains relatively acceptable despite this progression towards harder water.

In the upper reaches of the Shule River, the $Na^+\%$ value in river water ranges from 6.05 to 18.69, with an average value of 14.25. This range categorizes the water as excellent, making it suitable for irrigation. In the middle reaches, among the five samples, only one exceeds a $Na^+\%$ value of 20 (20.48), while the remaining samples also qualify as excellent for irrigation. Downstream, the $Na^+\%$ value in river water samples varies from 16.3 to 59.17, with an average of 29.18. Of the eight samples in the downstream area, 25% are considered excellent, 12.5% fall into the permissible range, and the rest maintain a good quality rating. Conversely, all groundwater samples meet the criteria for excellent quality according to the $Na^+\%$ value. This implies that both river water and groundwater in the SRB are well-suited for irrigation purposes.

The concentration of nitrates in water often serves as a pivotal indicator for assessing the potential influence of human activities on aquatic ecosystems. This is particularly relevant due to the nitrate emissions associated with agricultural practices and livestock husbandry. In the study area, the nitrate concentration exhibited a range from 0.88 mg/L to 6.01 mg/L, with a mean

concentration of 3.32 mg/L. Notably, the nitrate levels in the waters of the Shule River consistently remained below the stipulated standards set by the World Health Organization for nitrate levels in potable water, which is 10 mg/L.

## 4.5 Implications for the regional hydrological cycle processes

The SRB, situated in the arid northwest region of China, stands out due to its limited precipitation and heightened evaporation. By integrating the isotopic signatures of various water sources involved in the hydrological cycle, including precipitation, glacial meltwater, river water, and groundwater, the spatial heterogeneity of groundwater and surface water interactions at the basin scale can be clearly characterized.

In the upper reaches of the Shule River, local precipitation represents the most important source of river recharge, while groundwater plays a critical role in maintaining the baseflow throughout the year. In this mountainous upstream region, both precipitation and glacier meltwater serve as important recharge sources for groundwater. Specifically, isotopic evidence indicates that groundwater in the Suli area is predominantly recharged by atmospheric precipitation, whereas groundwater in the Changma area is strongly influenced by glacier meltwater contributions. In contrast, in the middle reaches of the Shule River, under arid climatic conditions, river water experiences progressive evaporation and receives minimal recharge from local precipitation. Instead, groundwater discharges to the surface as springs, contributing significantly to river flow and serving as an additional major source alongside upstream inflow. In the lower reaches, the river water exhibits even more pronounced evaporative signatures. Under the influence of anthropogenic activities, including river water diversion for irrigation and groundwater abstraction, the interactions between groundwater and surface water become particularly close and dynamic. Therefore, a discernible hydraulic connection exists between river water and groundwater in the middle and lower reaches. From a hydrogeological perspective, in the vicinity of the Yumen alluvial fan, numerous springs overflow, ultimately feeding into the river again, constituting a zone where groundwater recharges surface water. Further downstream, in the Guazhou area, river water is extensively employed for agricultural irrigation and subsequently infiltrates, recharging groundwater. Consequently, in the middle and lower reaches of the SRB, there is a close relationship between groundwater and surface water, marked by frequent exchanges (Fig. 11).

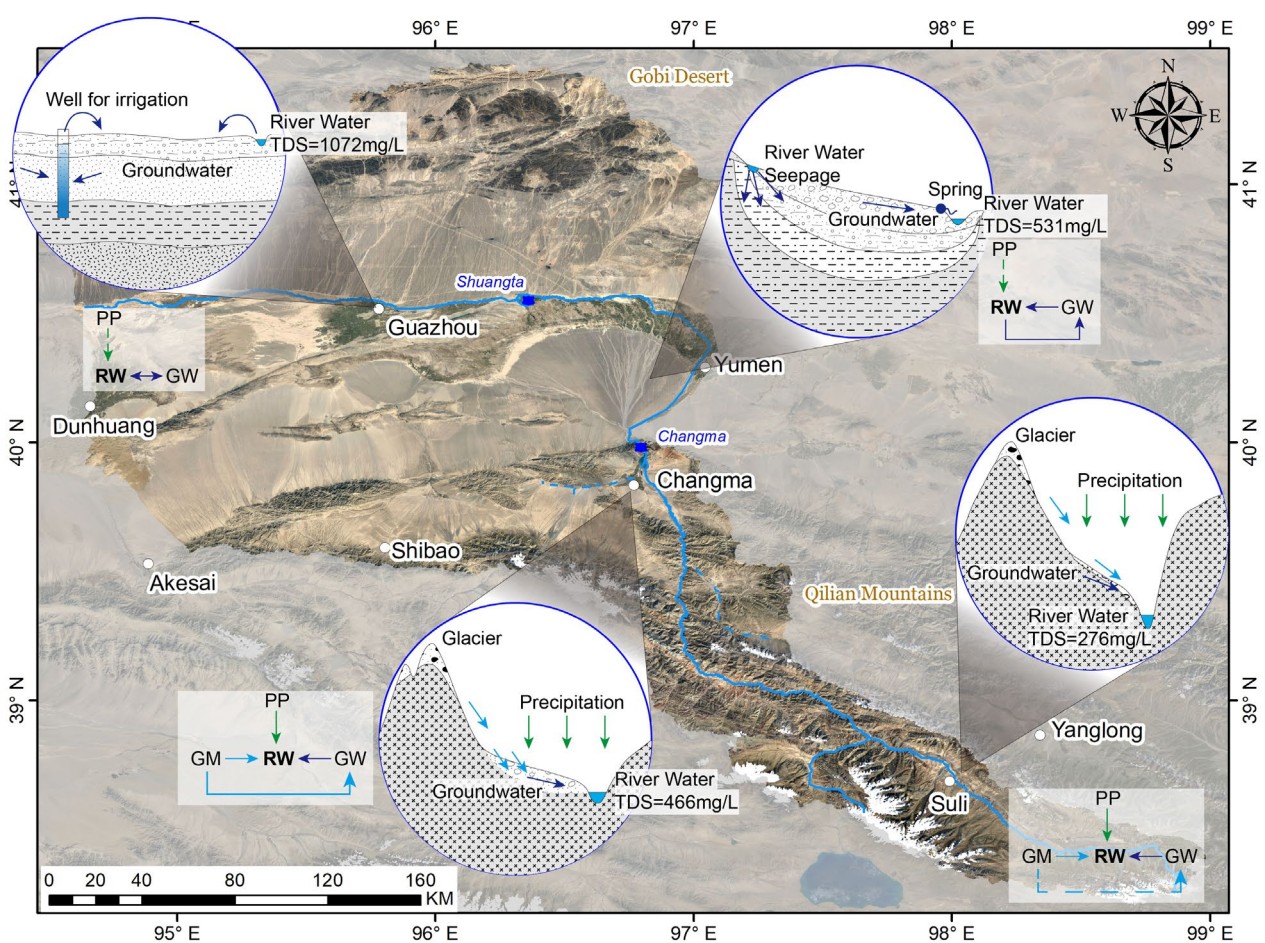

**Figure 11: Conceptual representation of basin-scale spatial heterogeneity in the hydrological cycle and associated water quality in the Shule River Basin. The base map is derived from satellite imagery of the SRB (Source: © Google Earth, accessed January 2025, Google Inc.). PP: precipitation; RW: river water; GW: groundwater; GM: glacier meltwater.**

Furthermore, combined with the hydrochemical features of river water and groundwater, the factors influencing regional water resource quality can be illustrated. The chemical composition of water in the Shule River is influenced by processes such as rock chemical weathering, including silicates and carbonates. Furthermore, evaporation also plays a non-negligible role in the hydrochemical features of river water. The chemical makeup of groundwater, on the other hand, is primarily controlled by water-rock interactions, involving mechanisms such as mineral dissolution and ion exchange. Both river water and groundwater are suitable for agricultural irrigation; however, in terms of drinking water quality, they are considered to be slightly hard. This not only establishes a scientific foundation for the prudent allocation and sustainable development of local water resources but also serves as a basis for the effective management of water resources and the enhancement of drinking water quality.

# 5 Conclusions

As a crucial agricultural oasis and a representative river-groundwater system in arid inland China, the SRB faces increasing challenges of water scarcity. This study integrates stable isotope and hydrochemical analyses to investigate the spatial heterogeneity of interactions between river water and groundwater at the basin scale. A conceptual model of the hydrological cycle was developed to provide a scientific basis for the sustainable development and utilization of regional water resources. The isotopic results reveal that the $\delta^{18}O$ value in Shule River water exhibit a clear altitude effect, with a gradient of $-0.08/100m$. This value is lower than those reported for the Qinghai-Tibet Plateau and Hexi Corridor. In the upper reaches, river water is mainly recharged by precipitation in the Qilian Mountains, with additional inputs from glacier meltwater and groundwater. Notably, glacier meltwater also contributes significantly to groundwater recharge in this region. In the middle reaches, after the river exits the mountains, river water infiltrates the riverbed at higher elevations, thereby recharging the groundwater. Conversely, at lower elevations near the Yumen alluvial fan, groundwater discharges as springs, which subsequently rejoin the river as a secondary recharge source. In the lower reaches, irrigation practices involving diverted river water result in a return flow that recharges the underlying groundwater. Due to minimal precipitation in both the middle and lower reaches, the contribution of direct rainfall to river or groundwater recharge is negligible. Overall, GW–SW interactions across the SRB are spatially heterogeneous and characterized by frequent bidirectional exchanges.

Chemical weathering processes of silicate and carbonate minerals significantly influence the chemical composition of river water, while dissolution and cation exchange reactions play crucial roles in shaping the hydrochemical composition of groundwater. In the upper reaches, where evaporation is limited and anthropogenic influence is minimal, both river water and groundwater exhibit low salinity, with average TDS values of 371.40 mg/L and 506.51 mg/L, respectively, indicating excellent water quality suitable for drinking and irrigation. In the middle reaches, TDS values increase moderately to 531.46 mg/L in river water and 506.85 mg/L in groundwater, suggesting water quality that remains suitable for irrigation. However, in the lower reaches, the combined effects of long-distance flow, intense evaporation under arid conditions, and heightened human activity contribute to notable degradation in water quality. Here, the average TDS values rise to 1072.13 mg/L in river water and 1499.65 mg/L in groundwater.

Understanding the interactions among various water sources in arid regions is essential for the rational allocation and sustainable management of water resources. However, substantial isotopic fractionation caused by strong evaporation presents a challenge for accurately quantifying water sources transitions. Future research should therefore focus on employing more conservative and evaporation-resistant tracers to improve the precision of source identification and mixing analysis. Such efforts are not only crucial for advancing the theoretical framework of environmental and hydrology, but also provide a scientific basis for the practical management and sustainable utilization of regional water resources in arid environments.

## Data availability

All data is available in Table S1. Map data can be downloaded from the United States Geological Survey.

## Author contribution

Conceptualization: LW and YD; Funding acquisition: YD; Investigation: LW and YS; Resources: CY; Visualization: LW and CY; Writing – original draft: LW and CY, Writing -review and editing: YD.

## Competing interests

The authors declare that they have no conflict of interest.

## Financial support

This research has been supported by Beijing Natural Science Foundation (Grant number. 8232049) and the National Natural Science Foundation of China (Grant number: 42474174 and 41702273).

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
