# Peer review of "Groundwater-Surface Water Interactions across an Arid River Basin: Spatial Patterns Revealed by Stable Isotopes and Hydrochemistry"

_EGUsphere, 2025_

## Author Response (AR1)

**Response to Comments**

Ms. Ref. No.: EGUSPHERE-2025-552

Dear Editor, Reviewers, and Community Commenter,

Thank you very much for your careful reading of our manuscript and for the constructive comments you have provided. Your insights have been invaluable in helping us to improve the quality and clarity of our work. We have addressed each point raised in a detailed, point-by-point response: all reviewer comments are shown in black font, and our replies in blue. Because the manuscript has undergone extensive reorganization, original line numbers could not always be retained; nevertheless, we have made every effort to refer to specific sections clearly. We greatly appreciate the time and effort you have invested in reviewing our submission.

**The editorial support team:**

1.   Please note that your reference list has not been compiled according to our standards. Please consider adjusting your reference list with the next revision of your manuscript. The manuscript preparation guidelines can be seen at: *https://www.hydrology-and-earth-system-sciences.net/for_authors/manuscript_preparation.html*.

**RESPONSE:AGREE AND CHANGES MADE**

Thank you for alerting us to the formatting inconsistencies in our previous reference list. We have carefully reformatted every citation in accordance with the journal's Author Guidelines. Please let us know if any further adjustments are required.

2.   For Figure 10 image credit, please use the proper attribution for Google Earth as described at *https://about.google/brand-resource-center/products-and-services/geo-guidelines/#required-attribution*

**RESPONSE:AGREE AND CHANGES MADE**

Thank you for bringing this to our attention; Figure 10 has been revised to comply with Google's required attribution guidelines.

**Reviewer #1:**

This manuscript presents a valuable investigation of groundwater-surface water interactions in an arid region basin using stable isotopes and hydrochemical analysis. The study appears methodologically sound and addresses an important gap in understanding hydrological processes in water-scarce environments. Although the description of river and groundwater dynamics in an arid region, using tools such as isotopes and hydrogeochemistry, but they are detailed analyzed.

Below I provide specific comments organized by manuscript section.

**Abstract:**

1. L10:China's Northwest region (e.g., Xinjiang, Gansu) shares some ethnic, linguistic, religious, and geographical similarities with Central Asia, but it cannot fully represent Central Asia.

**RESPONSE:AGREE AND CHANGES MADE**

Thank you for your valuable comment. You are absolutely correct that while China's northwest region (e.g., Xinjiang, Gansu) shares certain ethnic, linguistic, religious, and geographical characteristics with Central Asia, it does not fully represent the entire region. We acknowledge that we have broadened the scope regarding Central Asia in the manuscript. We will revise this statement to ensure greater accuracy and clarity.

2. The paper is entitled "Interactions of Surface Water and Groundwater", yet the abstract fails to elaborate on the mechanisms of their interaction.

**RESPONSE:AGREE AND CHANGES MADE**

Thank you for your comment. We have revised the abstract in the updated manuscript to clarify the mechanisms of surface water and groundwater interactions, ensuring consistency with the title.

3. L14: Isotope notation must be formatted as superscript, such as: $\delta^{18}O$.

**RESPONSE:AGREE AND CHANGES MADE**

Thank you for your comment. We will correct this issue and conduct a thorough review of the manuscript to ensure that similar errors do not occur elsewhere.

4. Consider adding a sentence about the practical implications of the findings for water resource management in arid regions.

**RESPONSE:AGREE AND CHANGES MADE**

Thank you for your comment. We have thoroughly revised the abstract and added a sentence highlighting the practical implications of our findings for water resource management in arid regions.

**Introduction**

5. Your introduction is very general and elaborate. The literature review adequately covers relevant studies but could better highlight the novelty of this work. The rationale for focusing on arid regions is not enough.

**RESPONSE:AGREE AND CHANGES MADE**

Thank you for your comment. We have revised the first, second, and fourth paragraphs of the introduction. First, we re-reviewed the latest published literature on groundwater–surface water interactions in arid regions and provided a detailed summary of these studies. Second, we have emphasized the significance and innovative aspects of our work. Finally, we clearly stated in the objectives that this study focuses on the heterogeneity of groundwater–surface water interactions at the watershed scale.

**Materials and methods**

6. One sampling campaign with merely 23 samples cannot adequately elucidate the groundwater-surface water interaction mechanisms.

**RESPONSE:AGREE AND CHANGES MADE**

Thank you for your comment. This issue is indeed crucial and has also been raised in community comments. The Shule River spans over 620 kilometers, from which we collected 23 surface water samples and 8 groundwater samples. Our sampling strategy was developed based on a thorough literature review and an understanding of the study area's specific characteristics. On one hand, as illustrated in Figure 4, the hydrochemical composition in the upper and mid-stream regions remains relatively consistent; significant changes are observed only in the downstream areas, such as near Shuangta Reservoir and on the Guazhou Plain. Thus, we believe that the 23 surface water samples are sufficiently representative. To further substantiate our approach, we have compiled additional published data on the Shule River and provided this information in Supplementary Table S1.

7.  L159: Well water samples are obtained from agricultural irrigation wells. Maybe the groundwater sample is mixed with different deep groundwater.

**RESPONSE:AGREE AND CHANGES MADE**

Indeed, this is true. Agricultural irrigation wells typically do not have screened intervals isolated by depth in order to maximize water extraction rates. We have acknowledged and clarified this limitation in the sample collection section of our manuscript, explaining the representative nature of the groundwater samples obtained.

8.  L160-164: What is the distance between the well and the river?

**RESPONSE:AGREE AND CHANGES MADE**

The distance between the wells and the river ranges from 0.5 to 9 km, with most wells located around 1 km away. We will include these details in the manuscript to further clarify the representativeness of our groundwater samples.

**Discussion**

9.  The sections 4.2.1, 4.2.2, 4.2.3 are results not discussion.

**RESPONSE:AGREE AND CHANGES MADE**

Thank you for your suggestion. We have reorganized these sections accordingly, ensuring that the relevant results have been moved to section 3.

10. Discussion needs to be improved to explain the relevancy of these findings. This is rather short in the current version.

**RESPONSE:AGREE AND CHANGES MADE**

Thank you for your suggestion. We have expanded the discussion section (Section 4) to more thoroughly explain and discuss the relevance of our findings.

11. Consider adding a conceptual model summarizing interactions.

**RESPONSE:AGREE AND CHANGES MADE**

Thank you for your suggestion. We have now included an explanation and summary of the conceptual model in both the discussion and conclusion sections of the manuscript.

**Reviewer #2:**

The manuscript submitted by Wang et al. presents a highly valuable contribution to the current frontiers in hydrology by elucidating the interactions between groundwater and surface water along a complete watershed in Central Asia. By integrating hydrochemical and isotopic techniques, the study effectively demonstrates the transformation process from river source areas to drainage zones and reveals the spatial heterogeneity of groundwater–surface water interactions. This work carries significant importance for understanding the hydrological processes across mountainous and plain regions. Overall, this manuscript is of considerable value to the field of hydrology, offering novel insights into the dynamics of groundwater and surface water interactions across diverse terrains. Thus, I recommend its publication following revisions.

However, some improvements in presentation and clarity will further enhance the manuscript. My detailed comments are as follows:

1. Line 14: Please ensure that isotopes are expressed with superscripts (e.g., $^{18}O$) throughout the manuscript. A careful review is needed, as a similar issue appears on Line 32.

**RESPONSE:AGREE AND CHANGES MADE**

In response to the comment on Line 14 (and the similar instance on Line 32), we apologize for the formatting oversight in our isotope notation. We have carefully reviewed the entire manuscript and corrected all instances to ensure that isotopes (e.g., $^{18}O$) are uniformly expressed with superscripts. Thank you for bringing this to our attention.

2. Line 267: There is a spelling error that should be corrected.

**RESPONSE:AGREE AND CHANGES MADE**

We sincerely apologize for the noted spelling error. We have carefully proofread

the entire manuscript and corrected this mistake, as well as any similar errors, to ensure the highest level of accuracy. Thank you for bringing this to our attention.

3. Line 184: Spelling error.

**RESPONSE:AGREE AND CHANGES MADE**

It has been corrected.

4. Line 38: One of the references is missing the publication year; please verify and amend this accordingly.

**RESPONSE:AGREE AND CHANGES MADE**

The publication year has been added.

5. Line 81-82: Consider including a reference regarding the number of glaciers mentioned.

**RESPONSE:AGREE AND CHANGES MADE**

Thank you for your comment and we have added references.

6. Line 430-431: It is recommended to add the appropriate references to support the statements made in these lines.

**RESPONSE:AGREE AND CHANGES MADE**

The appropriate references have been added.

7. Lines 50-52: I suggest that the authors further elaborate on the significance and innovative aspects of their study. After reviewing the current state of research in the introduction, stressing the distinctive contributions of the present study would greatly enhance its impact and facilitate broader dissemination within the community.

**RESPONSE:AGREE AND CHANGES MADE**

In response to this insightful suggestion, we have revised the Abstract to more explicitly articulate the significance and novelty of our work. Specifically, we now

highlight how our study advances current understanding by discussing the spatial heterogeneity of groundwater–surface water interactions—from mountainous to plain regions—at the watershed scale in arid environments, and by introducing a novel application of isotopic and hydrochemical methods across multiple water bodies, including groundwater, glacier meltwater, and river water, in the typical arid areas. We believe these enhancements clearly underscore the distinctive contributions of our research and will facilitate broader dissemination within the hydrological community. Thank you for encouraging us to strengthen this aspect of the manuscript.

8.  Lines 70-75: Given the focus on the topic and the subsequent conclusions, please incorporate a statement in the study objectives regarding the construction of a conceptual model to explain the spatial heterogeneity of groundwater–surface water interactions.

**RESPONSE:AGREE AND CHANGES MADE**

We have amended both the Abstract and the Conclusions to explicitly include, as a key study objective, the construction of a conceptual model aimed at explaining the spatial heterogeneity of groundwater–surface water interactions. This addition clarifies how our conceptual framework integrates findings across mountainous and plain regions, thereby strengthening the link between our data and the overarching watershed-scale patterns. Thank you for prompting us to make this enhancement

9.  Lines 105-110: I recommend including a brief introduction on the intra-annual variations in precipitation. This information could help readers better understand the hydrological cycle characteristics in arid regions.

**RESPONSE:AGREE AND CHANGES MADE**

It has been added to the current manuscript.

10. Line 119: Please provide the data source for Figure 2.

**RESPONSE:AGREE AND CHANGES MADE**

The data source has been added to the Figure Caption.

11. Line 178: It is advisable to add the computational method for d-excess.

**RESPONSE:AGREE AND CHANGES MADE**

It has been added.

12. Line 190: When discussing Changma Reservoir and Shuangta Reservoir, please refer the reader to Figure 1 for better geographic orientation.

**RESPONSE:AGREE AND CHANGES MADE**

Thank you for your comment and we have revised the manuscript.

13. Line 96-98: It is advisable to add details regarding the base map and the source of the elevation data for Figure 1. In addition, it is suggested that authors mark the blue part in Figure 1a as the SRB.

**RESPONSE:AGREE AND CHANGES MADE**

Thank you for your suggestion and we have revised them.

14. Line 235: The title of Figure 6 appears to be incorrect. Moreover, consider relocating Figure 6 to the discussion section to better integrate and illustrate the findings.

**RESPONSE:AGREE AND CHANGES MADE**

We are very sorry that we made such a mistake when editing the manuscript to HESS format. We have checked and corrected all the errors. Thank you for your suggestion.

15. Line 365: The title of Figure 9 contains an error; a comprehensive review of all figure titles is recommended to ensure they are correct.

**RESPONSE:AGREE AND CHANGES MADE**

It has been corrected.

16. Section 4.2: It is advisable to reposition the discussion of the Gibbs diagram

and ionic ratios to the results section. In addition, the text between Lines 287–295 is somewhat repetitive and redundant; please consider rewriting this portion for enhanced clarity.

**RESPONSE:AGREE AND CHANGES MADE**

We have reorganized the manuscript to improve its logical flow and readability. Specifically, the discussion of the Gibbs diagram and ionic ratios has been moved from the Discussion to the Results section. We have also rewritten the text originally between Lines 287–295 to remove redundancy and enhance clarity. Thank you for helping us strengthen the presentation of our findings

17. Line 410: I suggest expanding Section 4.3.2 to more comprehensively describe the hydrochemical evolution process including river water and groundwater.

**RESPONSE:AGREE AND CHANGES MADE**

We have substantially expanded Section 4.3.2 to offer a more comprehensive account of the hydrochemical evolution processes affecting both river water and groundwater. In the revised manuscript, we now: 1) Detail the sequential changes in major ion concentrations and isotopic signatures from upstream headwaters to downstream alluvial plains. 2) Describe the mixing dynamics between surface and subsurface flows, highlighting key zones of geochemical interaction. We trust that these enhancements provide a clearer and more thorough understanding of the hydrochemical evolution across the study area. Thank you for prompting us to enrich this section.

18. Section 4.4: Please include a reference for the water quality evaluation standards. Additionally, add the calculations for TH, Na%, and SAR in a supplemental table.

**RESPONSE:AGREE AND CHANGES MADE**

In response to this comment, we have added the appropriate citation for the water quality evaluation standards (WHO, 2010) in the Methods section. Furthermore, we

have included a new Supplemental Table S1 detailing the calculations of Total Hardness (TH), Sodium Percentage (Na %), and Sodium Adsorption Ratio (SAR) for all sampling sites. Thank you for this helpful suggestion.

*WHO (2010) Hardness in drinking-water: background document for development of WHO guidelines for drinking-water quality. World Health Organization, Geneva*

19. Lines 250-255: The authors should further clarify how the LMWL are established and elaborate on the rationale for developing different LMWL. This additional explanation would assist readers in comprehending the underlying assumptions and methodology.

**RESPONSE:AGREE AND CHANGES MADE**

We have expanded the manuscript to provide a more detailed account of how the LMWL are established and why separate lines are developed. Specifically, we now: 1) Describe the precipitation sampling network and isotopic dataset used to derive each LMWL. 2) Outline the least-squares regression approach applied to $\delta^{18}O$ versus $\delta^2H$ data to generate the line parameters. 3) Clarify the criteria for defining distinct sub-regional LMWLs—namely, how variations in climate regime and isotopic composition justify separate lines to ensure each is appropriately applicable. These additions elucidate the underlying assumptions, data sources, and methodological rationale for developing different LMWLs. Thank you for encouraging us to make this improvement.

20. In the conclusions section, it would be beneficial to explicitly address all the study objectives mentioned in the introduction. Please ensure that the spatial heterogeneity of groundwater–surface water interactions is clearly emphasized. Furthermore, discuss the implications of these findings for similar regions facing analogous issues and for the sustainable management of regional water resources.

**RESPONSE:AGREE AND CHANGES MADE**

In response to this valuable suggestion, we have revised the Conclusions section

to explicitly address all the study objectives outlined in the Introduction. In particular, we now place stronger emphasis on the spatial heterogeneity of groundwater–surface water interactions across different geomorphic zones of the watershed. Additionally, we discuss the broader implications of our findings for arid and semi-arid regions facing similar hydrological challenges, and highlight how the insights gained from our conceptual model and hydrochemical analyses can inform the sustainable management and allocation of regional water resources. We appreciate your thoughtful recommendation, which has helped us to strengthen the overall impact and relevance of our conclusions.

**CC1:**

It is a great work, it is worth publishing.

1. The introduction effectively highlights the importance of studying groundwater-surface water interactions in arid regions. However, the authors should elaborate on the existing knowledge gaps that this study aims to fill.

**RESPONSE:AGREE AND CHANGES MADE**

Thank you very much for your recognition and valuable suggestion. We have rewritten the introduction as per your recommendation and that of Reviewer 1. The revised introduction now explicitly details the existing knowledge gaps that our study aims to address. Thank you again for your helpful comments.

2. The discussion of previous studies (e.g., Zhou et al., 2015; Wang et al., 2016) is useful but could be more critically analyzed to highlight the limitations of past work and justify the need for this study.

**RESPONSE:AGREE AND CHANGES MADE**

Thank you for your valuable suggestion. You are absolutely correct that while some studies have been conducted in this watershed, previous work has largely focused on small-scale analyses of groundwater recharge sources and evolution processes. Inspired by these earlier investigations, our study aims to address groundwater–surface water interactions at the entire watershed scale, providing a more comprehensive and holistic perspective. In response to your comment, we have added a paragraph in the discussion section that critically assesses the limitations of earlier studies and outlines the novelty and strengths of our work. Thank you again for your insightful feedback.

3. The sample size (31 samples) is relatively small for a large-scale basin. The authors should discuss whether this sample size is sufficient for robust conclusions.

**RESPONSE:AGREE AND CHANGES MADE**

Thank you for your comment. You are absolutely correct that a sample size of 31 for such a large-scale basin may initially seem limited. We have now incorporated additional analysis in the discussion section to address the representativeness of our samples and to justify the robustness of our conclusions. Thank you again for your valuable feedback.

4. The identification of groundwater recharge sources is well-explained, but there is no attempt to quantify the contribution of different sources using mixing models or statistical analysis.

**RESPONSE:AGREE**

Thank you very much for your valuable comment. Indeed, applying an end-member mixing model to quantify the contributions of different recharge sources is a critical approach. However, previous studies have attempted to address this issue, and it is important to note that in a large-scale basin, groundwater recharge sources can vary significantly, making it challenging to define consistent end-members. Furthermore, such an analysis might divert the focus from the primary objectives of our study. For these reasons, we have chosen to refrain from incorporating a quantitative contribution analysis in this manuscript. Thank you again for your thoughtful suggestion.

5. The role of fault structures and subsurface geological formations in influencing groundwater flow and interaction with surface water should be elaborated.

**RESPONSE:AGREE AND CHANGES MADE**

Thank you for your valuable suggestion. Indeed, fault structures play an indispensable role in controlling groundwater flow and its interactions with surface water, especially in transitions from mountainous regions to basins. We have now added content related to this aspect in the Materials and Methods section. Thank you for your helpful feedback.

6. I strongly recommend to mention other method in the methodology for SW-GW interaction and cite papers such as: "Assimilation of Sentinel-Based Leaf Area Index for Modeling Surface-Ground Water Interactions in Irrigation Districts".

**RESPONSE:AGREE AND CHANGES MADE**

Thank you for your valuable suggestion. I have noted the important and innovative paper you mentioned. In response, we have revised the introduction in the updated manuscript to comprehensively review various methods used to study groundwater–surface water interactions, and we have incorporated relevant citations including the suggested paper. Thank you again for your helpful recommendation.

*Zafarmomen, N., Alizadeh, H., Bayat, M., Ehtiat, M., and Moradkhani, H.: Assimilation of Sentinel-Based Leaf Area Index for Modeling Surface-Ground Water Interactions in Irrigation Districts, Water Resour. Res., 60, e2023WR036080, https://doi.org/10.1029/2023WR036080, 2024.*

---

## Author Response (AR2)

**Response to Comments**

Ms. Ref. No.: EGUSPHERE-2025-552

Dear Editor and Reviewers,

We would like to sincerely thank the editor and all reviewers for their valuable time, constructive comments, and insightful suggestions. We are especially grateful to the reviewers from the first round for their detailed and specific feedback, which greatly helped us improve the quality and clarity of the manuscript. We also appreciate the continued support and thoughtful input from the editor and reviewers during the second round of review. Their efforts and professional insights are deeply appreciated and have been instrumental in refining our work.

We have carefully considered each comment and revised the manuscript accordingly. Below, we provide a detailed, point-by-point response, with reviewer comments in black and our responses in blue for clarity.

**Reviewer #1:**

1.  86–88: Present the contribution in a "local-to-global" sequence: first explain the value for the Shule River Basin itself, then broaden to analogous arid-zone basins worldwide. This progression makes the argument more reader-friendly.

**RESPONSE:AGREE AND CHANGES MADE**

We greatly appreciate this important suggestion. Following your advice, we have modified the relevant sentence to follow a clearer "local-to-global" logic, which improves the readability and flow of the argument. Please see lines 86–88 in the revised manuscript.

2.  Line 100 :he reported mean altitude of the basin does not feed into any subsequent analysis or discussion. Consider deleting the sentence to keep the background concise.

**RESPONSE:AGREE AND CHANGES MADE**

We appreciate your suggestion. To improve the conciseness of the background section, we have removed the sentence as recommended

3.  Isotopic notation consistency: Wherever hydrogen and oxygen isotopes are cited, use the fixed order "δD and δ18O" (e.g., "the δD and δ18O signatures indicate …"). A quick search shows a few instances where the order is reversed—unify these for stylistic consistency.

**RESPONSE:AGREE AND CHANGES MADE**

We greatly appreciate your insightful comment. We have thoroughly reviewed the manuscript and unified the isotopic notation to consistently use "$\delta D$ and $\delta^{18}O$" as recommended.